# 6-Phosphogluconate dehydrogenase (6PGD), a key checkpoint in reprogramming of regulatory T cells metabolism and function

Saeed Daneshmandi[1,2]*, Teresa Cassel[3], Richard M Higashi[3], Teresa W-M Fan[3]*, Pankaj Seth[1,2]*

[1]Department of Medicine, Beth Israel Deaconess Medical Center, Harvard Medical School, Boston, United States; [2]Division of Interdisciplinary Medicine, Beth Israel Deaconess Medical Center, Harvard Medical School, Boston, United States; [3]Center for Environmental and Systems Biochemistry, University of Kentucky, Lexington, United States

**Abstract** Cellular metabolism has key roles in T cells differentiation and function. $CD4^+$ T helper-1 (Th1), Th2, and Th17 subsets are highly glycolytic while regulatory T cells (Tregs) use glucose during expansion but rely on fatty acid oxidation for function. Upon uptake, glucose can enter pentose phosphate pathway (PPP) or be used in glycolysis. Here, we showed that blocking 6-phosphogluconate dehydrogenase (6PGD) in the oxidative PPP resulted in substantial reduction of Tregs suppressive function and shifts toward Th1, Th2, and Th17 phenotypes which led to the development of fetal inflammatory disorder in mice model. These in turn improved anti-tumor responses and worsened the outcomes of colitis model. Metabolically, 6PGD blocked Tregs showed improved glycolysis and enhanced non-oxidative PPP to support nucleotide biosynthesis. These results uncover critical role of 6PGD in modulating Tregs plasticity and function, which qualifies it as a novel metabolic checkpoint for immunotherapy applications.

*For correspondence:
Daneshmandi2006@yahoo.com (SD);
teresa.fan@uky.edu (TW-MF);
sethpankaj829@gmail.com (PS)

**Competing interest:** The authors declare that no competing interests exist.

## Introduction

Regulatory T cells (Tregs) are distinct subsets of $CD4^+$ T cells that limit inflammatory responses by suppressing self-reactive T cells to maintain balance between protective and excessive inflammation. Although lipid oxidation and the mevalonate pathway being considered key to their functions (*Michalek et al., 2011*; *Timilshina et al., 2019*), cellular metabolism in Tregs is an active area of investigation. Tregs depend on these pathways to maintain their suppressive functions, which are distinct from the high glycolytic signatures of conventional $CD4^+$ effector subsets, Th1, Th2, and Th17 (*Michalek et al., 2011*). Although glycolysis is not major feature in Tregs, it is critical for Treg expansion (*Pacella et al., 2018*). Glucose (Glc) enters the cell via glucose transporter 1 (Glut1) and is metabolized to glucose 6-phosphate (G6P), which has alternative fates via glycolysis and as the entry point into the pentose phosphate pathway (PPP) (*Figure 1—figure supplement 1*).

The PPP, also known as the hexose monophosphate shunt or phosphogluconate pathway, comprises two interdependent branches: the oxidative branch and the non-oxidative branch. Oxidative PPP is composed of three essentially irreversible reactions in which the first is catalyzed by G6PD that converts G6P to 6-phosphogluconolactone (6 PGL) while generating reduced nicotinamide adenine dinucleotide phosphate (NADPH). 6 PGL then hydrolyzes to 6-phosphogluconate (6 PG), which is oxidatively decarboxylated by 6-phosphogluconate dehydrogenase (6PGD), producing ribulose-5-phosphate

and a second molecule of NADPH and (*Figure 1—figure supplement 1A*). Ribulose-5-phosphate is the precursor for ribose-5-phophate (R5P) and thence nucleotides, and NADPH is required both for reductive anabolism and for antioxidative defense.

Tregs are known to have high rates of lipid oxidation (*Michalek et al., 2011*), which would provoke antioxidant machinery which requires NADPH generated from PPP. We thus investigated the role of PPP in T CD4+ subsets. Our preliminary studies using inhibitors of the PPP enzymes indicated that the third enzyme in the oxidative PPP, 6PGD, has critical roles in modulating T cell functions.

To assess the role of 6PGD in modulating Treg function, we generated mice with flox targeted *Pgd* allele (*Pgd*fl/fl), which were crossed with *Foxp3*YPF-Cre mice to generate *Pgd*fl/fl*Foxp3*YPF-Cre (thereafter *Pgd*fl/fl*Foxp3*Cre). Interestingly, *Pgd*fl/fl*Foxp3*Cre mice showed a distinct phenotype, namely spontaneous fatal autoimmunity and lower lifespan. Isolated *Pgd*-deficient Tregs had diminished suppressive functions both in vitro and in vivo. They also displayed shift in differentiation toward Th1, Th2, and Th17 CD4+ T cell subtypes. Stable isotope tracing studies on sorted Tregs revealed reprogrammed metabolic pathways consistent with the generation of these subsets. Our results for Tregs directly isolated from pro-inflammatory mice model and derived from naïve CD4+ T cells in vitro uncover a critical role of 6PGD in maintaining Tregs metabolism, phenotype, and function.

## Results

### PGD deficiency in Tregs induces spontaneous fatal autoimmunity

Our preliminary studies showed that pharmacological inhibitor of 6PGD (6-aminonicotinamide [6-AN]) but not G6PD inhibitor (dehydroepiandrosterone [DHEA]) enhances IFN-γ production by in vitro activated CD4+ T cells (*Figure 1—figure supplement 1B*), while DHEA but not 6-AN at 10 μM instigated poor T cells viability (*Figure 1—figure supplement 1C*). Based on these, we generated *Pgd*fl/fl mice (*Figure 1—figure supplement 1D*) and crossed with *Cd4*Cre mice (*Figure 1—figure supplement 1F*). An enhanced effector phenotype of T CD4+ subsets was observed in *Pgd*fl/fl*Cd4*Cre mice (*Figure 1—figure supplement 1I-J*) suggesting altered regulation of T cell differentiation and/or activation.

To specifically delete 6PGD in Tregs, *Pgd*fl/fl mice were crossed with *Foxp3*Cre mice (*Figure 1—figure supplement 1G*). In these mice, the Cre recombinase driven by the *Foxp3* promoter will delete *Pgd* only in Tregs upon *Foxp3* expression and YFP can be used as expression marker (*Rubtsov et al., 2008*). *Pgd* deletion in YFP expressing cells was verified by western blot, real-time PCR (*Figure 1A–B*, *Figure 1—figure supplement 1E*), and stable isotope-resolved metabolomics (SIRM) analysis . Notably, disruption of 6PGD in Tregs led to a profound inflammatory disorder while *Pgd*fl/fl*Cd4*Cre mice appeared normal (*Figure 1—figure supplement 1K-L*). Compared with *Foxp3*Cre (wild-type [WT]) controls at 20 days of age, *Pgd*fl/fl*Foxp3*Cre mice exhibited a reduced body size and crusting of ears and eyelids (*Figure 1C*), enlargement of peripheral lymphoid organs (*Figure 1D*), and lymphoproliferation (*Figure 1E*) that was accompanied by a significantly shorter lifespan (*Figure 1F*). Lymphoproliferation was attributed to a higher frequency of CD4+ and CD8+ T cells in spleen (*Figure 1G*) with increased frequencies of CD44highCD62Llow effector/memory phenotype (*Figure 1H*), reduced frequencies of CD44lowCD62Lhigh naïve cells (*Figure 1H*), higher expressions of CD69 activation marker (*Figure 1I*), and enhanced production of IFN-γ (*Figure 1J*) in both CD4+ and CD8+ T cells. In these mice, CD8+ T cells also produced more granzyme-B and higher cytotoxic marker CD107a expression (*Figure 1K*). The titers of serum antibodies were much higher in *Pgd*fl/fl*Foxp3*Cre than WT mice (*Figure 1L*). Moreover, *Pgd*fl/fl*Foxp3*Cre mice had a significant increase in serum levels of IFN-γ, IL-17A, IL-4, and IL-5 (*Figure 1M*). Along with splenomegaly and lymphadenopathy in *Pgd*fl/fl*Foxp3*Cre mice, there was an enhanced immune cell infiltration and tissue damage in main organs (*Figure 1N*). The development of observed fatal inflammatory disease in *Pgd*fl/fl*Foxp3*Cre mice suggests a crucial role of 6PGD in modulating the function of Tregs. Such phenotype has also been reported in association with central signaling elements of metabolic reprogramming such as mechanistic target of rapamycin (mTOR) (*Zeng et al., 2013*) and liver kinase B1 (*Yang et al., 2017*).

### Deletion of 6PGD in Tregs results in loss of their suppressive function

Based on the observed phenotype in *Pgd*fl/fl *Foxp3*Cre mice, we further evaluated specific phenotypic markers and suppressive functions of Tregs, which were either isolated from inflamed mice model or generated in vitro from naïve CD4+ T cells. The CD4+Foxp3+ cells in the spleen of 20 -day-old *Pgd*fl/

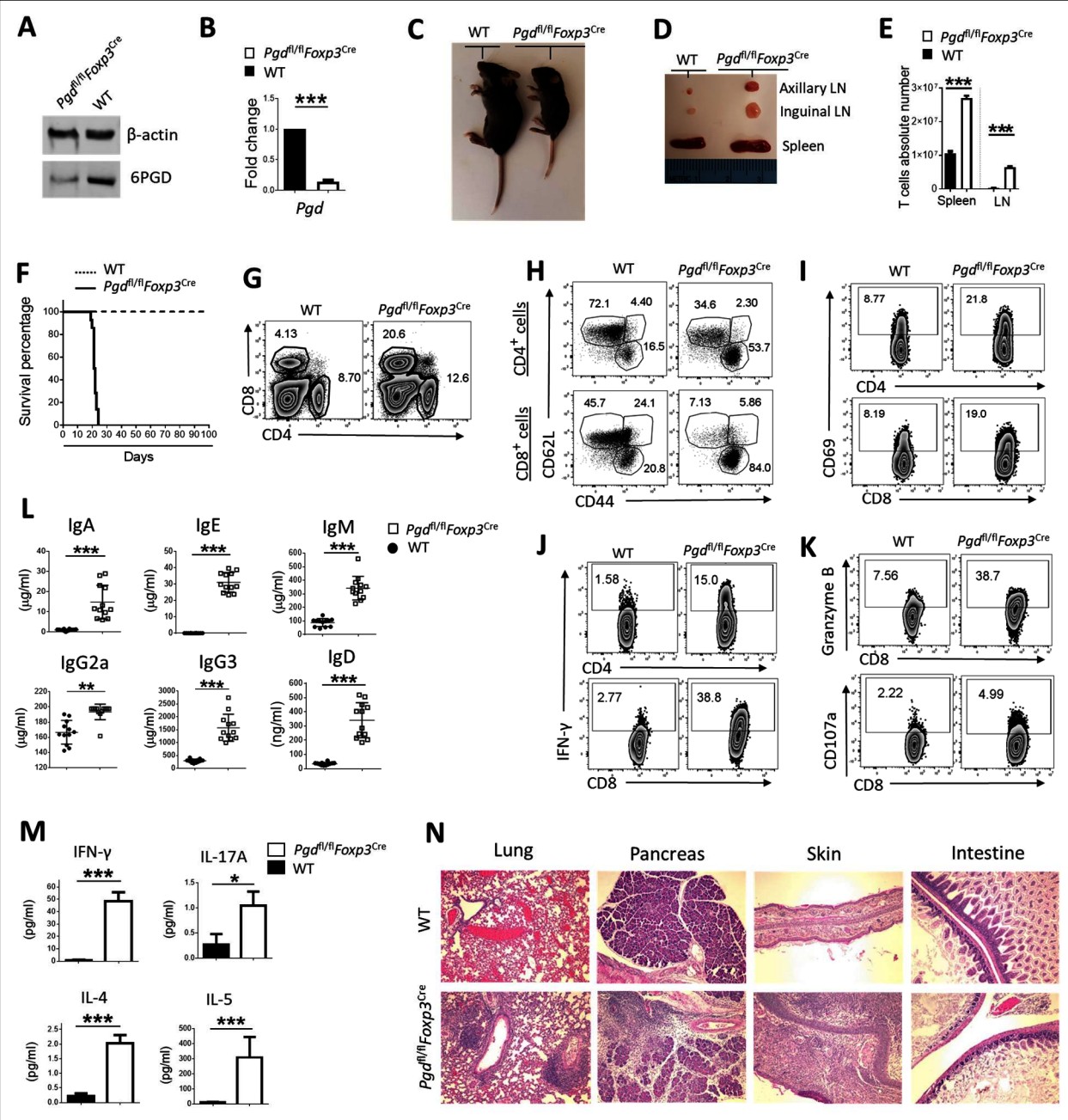

**Figure 1.** Deletion of 6-phosphogluconate dehydrogenase (6PGD) in regulatory T cells (Tregs) induces a fatal autoimmune phenotype. (**A–B**) YFP+ cells were sorted from *Pgd*+/+*Foxp3*Cre (wild-type [WT]) and *Pgd*fl/fl*Foxp3*Cre mice and deletion of *Pgd* was confirmed by western blot (**A**) and real-time PCR (**B**). (**C**) Representative image of 21 -day-old WT and *Pgd*fl/fl*Foxp3*Cre mice. (**D**) Representative image of lymphadenopathy in *Pgd*fl/fl*Foxp3*Cre compared to WT mice. (**E**) T cells absolute number per spleen and peripheral lymph nodes (pLNs) of WT and *Pgd*fl/fl*Foxp3*Cre mice are shown (N = 12 mice per group). (**F**) Survival curve of WT and *Pgd*fl/fl*Foxp3*Cre mice. Representation of 15 mice per group. (**G**) Splenocytes from 20 -day-old WT and *Pgd*fl/fl*Foxp3*Cre mice were harvested and distribution of CD4+ versus CD8+ T cells were evaluated. (**H**) Deletion of 6PGD induces enhanced effector phenotype (CD44high CD62Llow) both in CD4+ T cells (top panel) and in CD8+ T cells (bottom panel) in 20 -day-old *Pgd*fl/fl*Foxp3*Cre compared to WT mice. (**I**) Both CD4+ and CD8+ T cells express higher levels of CD69 activation marker in *Pgd*fl/fl*Foxp3*Cre compared to WT mice. (**J**) Splenocytes from 20 -day-old WT and *Pgd*fl/fl*Foxp3*Cre mice were stimulated with PMA (50 ng/ml)/ionomycin (1 µg/ml) plus GolgiPlug (1 µl/ml) for 4  hr and expression of IFN-γ  was evaluated by flow cytometry. (**K**) Granzyme B (top panel) and CD107a degranulation activation marker (bottom panel) expression was assessed on splenocytes of 20 -day-old WT and *Pgd*fl/fl*Foxp3*Cre mice by flow cytometry. (**L–M**) Serum was collected from 20 -day-old WT and *Pgd*fl/fl*Foxp3*Cre mice and levels of serum antibodies (**L**) and IFN-γ , IL-17A, IL-4, and IL-5 (**M**) were detected as described in supplemental information. Results are representative of 12 mice per group. (**N**) Hematoxylin and eosin staining of lung: original magnification (**X10**), pancreas (**X10**), skin (**X10**), and intestine (**X10**) of WT and *Pgd*fl/fl*Foxp3*Cre mice. *p < 0.05; **p < 0.01; ***p < 0.001.

*Figure 1 continued on next page*

*Figure 1 continued*

The online version of this article includes the following figure supplement(s) for figure 1:

**Figure supplement 1.** Effect of G6PD/6PGD inhibition and generation of *Pgd*<sup>fl/fl</sup> *Foxp3*<sup>cre</sup> mice.

<sup>fl</sup>*Foxp3*<sup>Cre</sup> mice were lower than the WT counterparts, both in number (**Figure 2A**) and in frequency of total CD4<sup>+</sup> T cells (**Figure 2B**). The expression of CD98 (a subunit of the neutral amino acid transporter), CD71 (the transferrin receptor), and CD40L in these 6PGD-deficient Tregs was enhanced (**Figure 2C**). CD71 and CD98 are key nutrient receptors in Tregs that depend on mTORC1 (**Kelly et al., 2007**).

Next, YFP<sup>+</sup> (indicator of Foxp3 expression) cells (**Rubtsov et al., 2008**) were sorted from 20 -day-old WT (*Foxp3*<sup>YFP-Cre</sup>) and *Pgd*<sup>fl/fl</sup>*Foxp3*<sup>Cre</sup> mice and evaluated for phenotypic and functional properties. YFP<sup>+</sup> sorted cells are demonstrated in **Figure 2D**. The level of Foxp3 transcript was lower in 6PGD-deficient cells evaluated by real-time PCR (**Figure 2E**), which is consistent with lower *Foxp3* gMFI in these cells (**Figure 2B**). When culturing sorted YFP<sup>+</sup> Tregs in vitro in the presence of IL-2 (700 IU/ml) and anti-CD3/anti-CD28 coated beads (Treg:beads ratio 1:3), higher number of cells were detected for 6PGD-deficient cells at both 24 and 48 hr time points (**Figure 2F**). This result showed that 6PGD deficiency was tolerable in Tregs and did not induce cell death in these cells. Evaluation of the culture media demonstrated significantly higher production of cytokines as markers for other CD4<sup>+</sup> T helper subsets such as IFN-γ (Th1), IL-13 and IL-5 (Th2), and IL-17A (Th17) due to 6PGD blockade (**Figure 2G**). RNA sequencing (RNAseq) analysis of isolated Tregs from inflamed mice also demonstrated higher expression levels of Th1 (*IFNg*, *Tbx21*), Th2 (*Il13*, *Il5*, *Il4* and *Gata3*), and Th17 (*Il17a*) marker genes in 6PGD-deficient Tregs, shown as heat map in **Figure 2—figure supplement 1A** and volcano plot in **Figure 2—figure supplement 1B**. RNAseq data were confirmed by real-time PCR for selected genes (**Figure 2—figure supplement 1C**) and flow cytometry (**Figure 2—figure supplement 1D**). It should be noted that although these shifts in gene expression are consequences of 6PGD blockade in Tregs, the resulting severe autoimmunity in mice with pro-inflammatory microenvironment should also be considered. Pro-inflammatory condition of mice could enhance or diminish some of these gene expression changes.

Although *Foxp3* expression is a key marker of the Treg formation, additional metabolic and transcriptional regulations can orchestrate Treg plasticity (**Shi and Chi, 2019**). *Pgd*-deficient Tregs can be converted to Th1, Th2, and Th17 by modification of metabolic pathways. Notably, the Th2 markers *IL13* and *IL5* were among the most upregulated genes in *Pgd* deleted Tregs (**Figure 2—figure supplement 1A-B**), which points to Treg differentiation into Th2 type cells. These gene expression changes were consistent with significantly reduced suppressive capacity of *Pgd*-deficient versus WT Treg (**Figure 2H–I**), which was assayed by co-culturing the corresponding isolated YFP<sup>+</sup> cells with CD4<sup>+</sup>CD45RB<sup>high</sup> effector cells in vitro, as described in supplemental information. Altered Treg function was along with enhancement of CD4<sup>+</sup> T cells phenotype shift toward Th1, Th2, and Th17 cells subsets in *Pgd*<sup>fl/fl</sup>*Foxp3*<sup>Cre</sup> mice (**Figure 2—figure supplement 1E-G**). Pro-inflammatory condition of mice could also contribute in the magnitude of these outcomes.

We also examined Tregs suppressive function in vivo using an inflammatory bowel disease (IBD) model (**Figure 2J**). YFP<sup>+</sup> cells from WT and *Pgd*<sup>fl/fl</sup>*Foxp3*<sup>Cre</sup> mice as well as WT CD4<sup>+</sup> T effector (CD4<sup>+</sup>CD45RB<sup>high</sup>) cells were isolated, mixed, and transferred to Rag1<sup>-/-</sup> mice (**Ostanin et al., 2009**; **Steinbach et al., 2015**). Higher rate of weight loss was observed in *Pgd*<sup>fl/fl</sup>*Foxp3*<sup>Cre</sup> group (**Figure 2K**) compared to WT. Colons were evaluated on day 45 post cells injection in the IBD model mice, 6PGD deficiency in Tregs led to shorter and thicker colons compared with no injection or injection with WT Treg (**Figure 2L**). Hematoxylin and eosin staining of colons demonstrated higher immune cell infiltration and destruction of colon structure (**Figure 2M**). Taken together these results confirm that 6PGD inhibition is associated with loss of Treg suppressive function, both in vitro and in vivo.

## 6PGD deficiency attenuates Treg suppressive function and induces potent anti-tumor responses

To confirm key observations from YFP<sup>+</sup> sorted cells of *Pgd*<sup>fl/fl</sup>*Foxp3*<sup>Cre</sup> mice that displayed inflammatory condition, we characterized changes in the molecular properties of WT Tregs induced by the small molecule inhibitor of 6PGD, 6-AN (**Davis and Kauffman, 1987**). We also examined the effect of

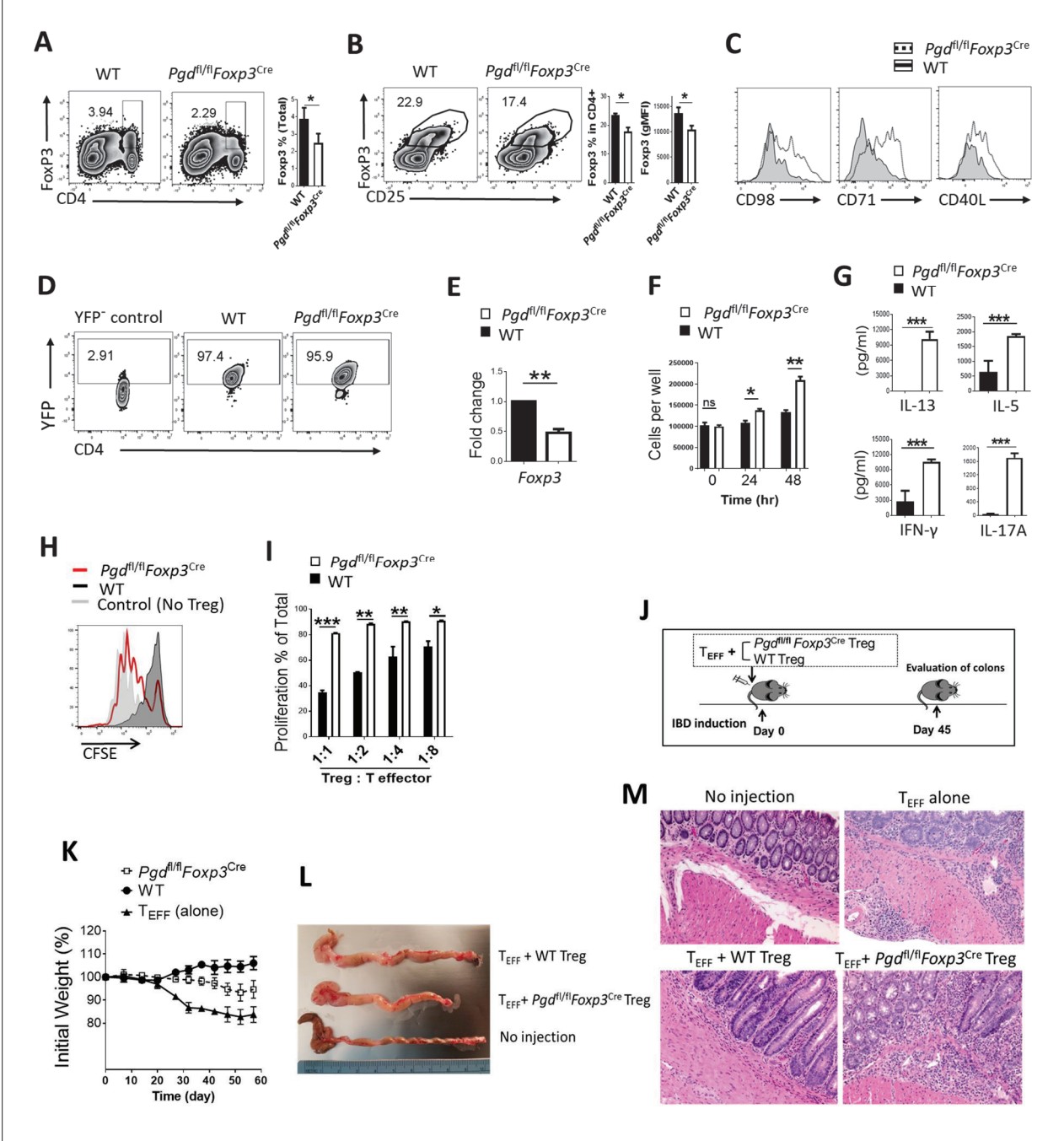

**Figure 2.** 6-Phosphogluconate dehydrogenase (6PGD) blockade results in loss of suppressive function in regulatory T cells (Tregs). (**A–B**) Frequency of CD4⁺Foxp3⁺ cells in spleen of 20- day-old *Pgd*⁺/⁺*Foxp3*ᶜʳᵉ (wild type [WT]) was evaluated and demonstrated (**A**) in total cells or (**B**) in gated CD4⁺ T cells. Bar graphs show respective statistical differences for percentage of Foxp3⁺ cells and Foxp3 gMFI. (**C**) Expression of CD98, CD71, and CD40L on Tregs from WT and *Pgd*ᶠˡ/ᶠˡ*Foxp3*ᶜʳᵉ mice. (**D–E**) YFP⁺ cells were isolated from 20 -day-old WT and *Pgd*ᶠˡ/ᶠˡ*Foxp3*ᶜʳᵉ mice. (**D**) Purified YFP⁺ cells are shown and (**E**) *Foxp3* mRNA levels were evaluated by real-time PCR. (**F**) Isolated Tregs (YFP⁺) were cultured in vitro in presence of IL-2 (700 IU/ml) and anti-CD3/anti-CD28 coaled beads (Treg:beads ratio 1:3) and cells number was assessed at 24 and 48 hr time points. Results are representative of three independent experiments with N = 4. (**G**) Cytokine release in media as cultured in above culture condition. (**H–I**) Isolated Tregs (YFP⁺) from *Pgd*ᶠˡ/ᶠˡ*Foxp3*ᶜʳᵉ mice showed lower suppressive activity in suppression assay, as described in Materials and methods. Representative histogram of CFSE dilution pattern (Treg:Teff = 1:1 ratio) (**H**) and bar graph of T cell proliferation as a function of serial Treg:Teff ratios (**I**) are shown. Results are from three independent experiments with N = 4. (**J**) Tregs (YFP⁺) from WT and *Pgd*ᶠˡ/ᶠˡ*Foxp3*ᶜʳᵉ mice and T effector (CD4⁺CD45RBʰⁱᵍʰ) cells were isolated, mixed, and transferred to *Rag1*⁻/⁻ mice as inflammatory bowel disease (IBD) model. Colons were evaluated 45 days after cells injection. (**K**) Weight change in IBD mouse model. (**L**) On day 45 post IBD induction *Rag1*⁻/⁻ mice colon were measured for length and thickness. (**M**) Representative hematoxylin and eosin staining of Rag1⁻/⁻

*Figure 2 continued on next page*

*Figure 2 continued*

mice colon on day 45 after IBD induction. Results are from two independent experiments with N = 8 per group. *p < 0.05; **p < 0.01; ***p < 0.001.

The online version of this article includes the following figure supplement(s) for figure 2:

**Figure supplement 1.** 6-Phosphogluconate dehydrogenase (6PGD) deficiency in regulatory T cells (Tregs) induces shift toward CD4$^+$ T helper subsets.

6PGD blockade in Tregs on tumor formation by tamoxifen treatment of *Pgd*$^{fl/fl}$ *Foxp3*$^{EGFP-Cre-ERT2}$ mice (*Figure 1—figure supplement 1H* and *Figure 3*).

First, we used WT *Foxp3*$^{Cre}$ naïve (CD62L$^{high}$ CD44$^{low}$) CD4$^+$ T cells to generate Tregs in vitro (as inducible Tregs [iTregs]) in the polarization medium (IL-2+ TGF-β) plus 6-AN or vehicle DMSO. This medium effectively induced *Foxp3* expression indicated by the YFP marker in both 6-AN and DMSO treatments (*Figure 3A*), with lower YFP gMFI in 6-AN treated cell (*Figure 3B*). Tregs generated under 6-AN treatment in vitro showed a lower level of *Foxp3* (*Figure 3C*), but a higher level of *Gata3* and *IL5* (*Figure 3C*) expression. This difference in *Foxp3/Gata3* expression in generated Tregs in vitro indicates altered Treg properties in 6PGD blocked cells. Tregs generated in the presence of 6-AN also showed diminished suppressive function in vitro (*Figure 3D–E*).

To control the timing of *Pgd* deletion in Tregs and evaluate Tregs suppressive function under controlled inflammation, we generated *Pgd*$^{fl/fl}$*Foxp3*$^{EGFP-Cre-ERT2}$ mice by crossing *Pgd*$^{fl/fl}$ mice with *Foxp3*$^{EGFP-Cre-ERT2}$ mice (*Figure 1—figure supplement 1H*). In these mice, expression of Cre in *Foxp3* promoter is induced by tamoxifen treatment and EGFP can be used as a detection marker (*Rubtsov et al., 2010*). Suppressive function of targeted Tregs was evaluated in vitro and in vivo in a tumor model. In 6 -week-old *Pgd*$^{fl/fl}$*Foxp3*$^{EGFP-Cre-ERT2}$ mice, tamoxifen was injected on days 0, 1, and 3 to induce deletion of 6PGD in Tregs. To induce tumor formation, B16F10 melanoma cells were injected into the flank region of the mice on day 10 post tamoxifen treatment (*Figure 3F*). Although the tamoxifen induction does not induce deletion in 100 % of Tregs in vivo, this model would induce sufficient 6PGD deletion in targeted Tregs to be tracked in vivo (*Rubtsov et al., 2010*). *Foxp3* expression was detected by *EGFP* expression (CD4$^+$CD25$^+$EGFP$^+$ cells in spleen) (*Figure 3G*). EGFP$^+$ sorted cells showed reduced suppressive activity in vitro (*Figure 3H*), as seen in the YFP$^+$ sorted cells from *Pgd*$^{fl/fl}$ *Foxp3*$^{Cre}$ mice (*Figure 2H–I*) or Tregs generated in presence of 6-AN (*Figure 3D–E*).

In the B16F10 tumor model, induced 6PGD deletion in Tregs resulted in reduced tumor growth (*Figure 3I*) and smaller final mass (*Figure 3J*) on day 16 post tumor induction. Assessing cell types in the tumor, we saw more CD8$^+$ and CD4$^+$ T cell infiltration (*Figure 3K–L*), along with higher IFN-γ production from both CD4$^+$ (*Figure 3M*) and CD8$^+$ (*Figure 3N*) T cells in *Pgd*$^{fl/fl}$*Foxp3*$^{EGFP-Cre-ERT2}$ versus WT mice. Thus, in the tumor microenvironment, Tregs were altered by 6PGD blockade to unleash host anti-tumor responses. As such, 6PGD can serve as a metabolic checkpoint for immune activation by modulating Tregs activity, which is considered to be desirable (*Wang et al., 2017*; *Pacella and Piconese, 2019*).

## 6PGD function in Tregs is required for control of allergic (Th2) responses

Tregs are responsible for suppression of T subset effector responses but they also have the plasticity to convert into these subsets (*Shi and Chi, 2019*). A well-defined organ to follow Th2 responses control by Tregs is lung. In *Pgd*$^{fl/fl}$*Foxp3*$^{Cre}$ mice, significant Th2 (allergic) responses were evident in the lung (*Figure 3—figure supplement 1*). These included higher collagen deposits (*Figure 3—figure supplement 1A*, top panels), higher mucus accumulation (*Figure 3—figure supplement 1A*, bottom panels), and higher infiltration of T cells (both CD4$^+$ and CD8$^+$ cells) in the lung (*Figure 3—figure supplement 1C*) with enhanced production of Th2 markers such as IL-13 (*Figure 3—figure supplement 1D*), IL-5 (*Figure 3—figure supplement 1F*), and IL-4 (*Figure 3—figure supplement 1H*) by CD4$^+$ T cells. The sorted YFP$^+$ Tregs from 6PGD blocked mice also expressed lower levels of *Foxp3* and higher levels of *Gata3* transcription factors (*Figure 3—figure supplement 1B*). Moreover, the Th2 markers were higher in the spleen of the *Pgd*$^{fl/fl}$*Foxp3*$^{Cre}$ mice (*Figure 3—figure supplement 1E,G,I*).

High allergic responses in *Pgd*$^{fl/fl}$ *Foxp3*$^{Cre}$ mice was also evident with higher infiltration of eosinophils (CD11c$^{low/-}$ Siglect-F$^+$) in the lung (*Figure 3—figure supplement 1J*; *Stevens et al., 2007*) and spleen (CD11b$^+$ Siglect-F$^+$) (*Figure 3—figure supplement 1K*; *Hey et al., 2015*). These allergic

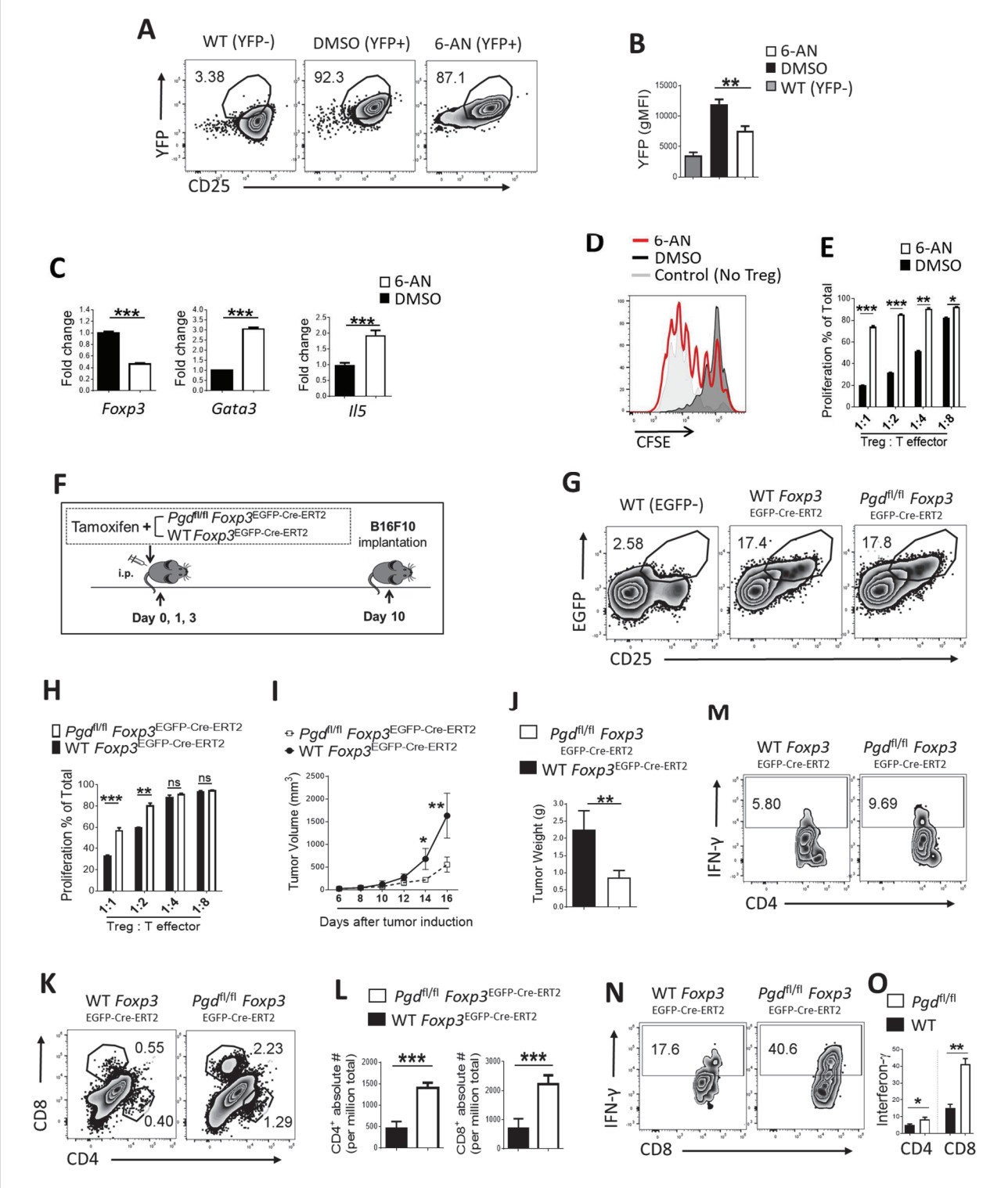

**Figure 3.** 6-Phosphogluconate dehydrogenase (6PGD) blockade in regulatory T cells (Tregs) prevents their suppressive function and induces potent anti-tumor responses. (**A–B**) Wild-type (WT) *Foxp3*[Cre] naïve CD4[+] T cells (CD62L[high] CD44[low]) were driven toward Tregs in vitro (as inducible Treg [iTreg]) along with treatment with 6-aminonicotinamide (6-AN) or vehicle DMSO and Treg drive efficacy was evaluated as CD25[+]YFP[+] cells. YFP[+] percentage in CD4[+] T cells (**A**) and YFP gMFI (**B**) are demonstrated. (**C**) Real-time PCR analysis on derived Tregs demonstrate lower expression of *Foxp3* transcription factor and higher expression of *Gata3* transcription factor and *IL-5* under treatment with 6-AN versus vehicle DMSO. (**D–E**) 6-AN treatment of driven Tregs demonstrate lower suppressive capacity evaluated in in vitro suppression assay. YFP[+] cells were sorted for the suppression assay. (**F**) Tamoxifen

*Figure 3 continued on next page*

**Figure 3 continued**

treatment schedule and tumor induction by implanting B16F10 cells in $Pgd^{fl/fl}$ $Foxp3^{EGFP-Cre-ERT2}$ and WT mice. (**G**) Flow cytometry analysis show the same frequency of Tregs (EGFP$^+$) in CD4$^+$ cells in spleen of both mouse groups 10 days post tamoxifen treatment. (**H**) Suppression assay on sorted EGFP$^+$ Tregs as described in supplemental information. (**I–J**) 6PGD blockade in Tregs resulted in lower tumor volume (**I**) and tumor weight (**J**). (**K–L**) Tumor of the mouse models from (**F**) were evaluated for infiltration of both CD4$^+$ and CD8$^+$ T cells on day 16 post implantation, which were higher in 6PGD blocked mice, as shown by representative flowgram (**K**) and bar graphs (**L**). (**M–O**) Tumor infiltrating CD4$^+$ (**M**) and CD8$^+$ (**N**) T cells showed higher capacity of IFN-$\gamma$ production on day 16. Results are from two independent experiments with N = 8 per group. *p < 0.05; **p < 0.01; ***p < 0.001.

The online version of this article includes the following figure supplement(s) for figure 3:

**Figure supplement 1.** 6-Phosphogluconate dehydrogenase (6PGD) function in regulatory T cells (Tregs) is required for suppressing allergic (Th2) responses.

responses were accompanied by infiltration of CD11b$^+$ Ly6G$^+$ cells in the lung and spleen of $Pgd^{fl/fl}Foxp3^{Cre}$ mice (**Figure 3—figure supplement 1L-M**; **Xue et al., 2020**; **Bronte et al., 2016**).

Although the shift toward Th2 responses (Figures: 1 M; **Figure 2—figure supplement 1B-C**; **Figure 3—figure supplement 1**; **Figure 2G**; and **Figure 3C**) was dominant in 6PGD-deficient Tregs, enhancement of Th1 and Th17 responses was also noted by higher IFN-γ and IL-17A in the serum (**Figure 1M**) and higher frequency of CD4$^+$IFN-γ$^+$ and CD4$^+$IL-17A$^+$ cells in the spleen (**Figure 2—figure supplement 1E, G**). Beside these enhanced Th1 and Th17 responses, YFP$^+$ Tregs isolated from $Pgd^{fl/fl}$ $Foxp3^{Cre}$ mice showed elevated expression of *Ifng, Tbx21,* and *Il17a* genes (**Figure 2—figure supplement 1A, C**) and sorted Tregs cultured in vitro secreted higher amount of IFN-γ and IL-17A into media (**Figure 2G**).

In other reports, comparable patterns of Treg plasticity have been observed by blocking a key regulator of Tregs metabolism, mTOR. $Mtor^{fl/fl}Foxp3^{Cre}$ mice showed 10- to 15-fold increases of Th2 mediators while increases in Th1 and Th17 responses were 2- to 3-fold (**Chapman et al., 2018**).

## 6PGD blockade in Tregs induces metabolic reprogramming similar to that of T effector cells

Next, we examined metabolic shifts in the Tregs isolated from WT and $Pgd^{fl/fl}Foxp3^{Cre}$ mice. It should be noted that the pro-inflammatory microenvironment of the mice could contribute to Tregs metabolic reprogramming, beside the intrinsic effects of 6PGD. We first assessed glycolysis and mitochondrial respiration by measuring the extracellular acidification rate (ECAR) and oxygen consumption rate (OCR) (**Figure 4A–B**). Tregs isolated from $Pgd^{fl/fl}Foxp3^{Cre}$ mice showed significant increase in both basal and reserve capacity in ECAR (**Figure 4A**) and OCR (**Figure 4B**) compared to WT Tregs. We also evaluated Glc uptake and mitochondrial status by fluorescent dye staining (**Figure 4C**). Compared to WT Tregs, 6PGD-deficient Tregs had increased Glc uptake capacity determined by 2-NBDG uptake (**Figure 4C**). These cells also showed increase in mitochondrial potential ($\Delta\Psi$m) as measured by TMRE (**Figure 4C**) without changes in the mitochondrial mass as examined by MitoTracker Deep Red FM (**Figure 4C**). Moreover, mitochondrial ROS (assessed by MitoSox Red) was enhanced in 6PGD blocked Tregs (**Figure 4C**). These results are consistent with the observed skew of the Tregs phenotype and markers toward conventional CD4$^+$ T effector subsets (**Michalek et al., 2011**; **Pacella et al., 2018**).

During transition from naïve CD4$^+$ T cells, Tregs switch from Glc utilization to other fuel sources such as fatty acids (**Procaccini et al., 2016**; **Macintyre et al., 2014**), which could help sustain their function in low Glc conditions such as in the tumor microenvironment. Glutamine (Gln) is an important fuel source for T cells that is consumed in mitochondria. However, it has been shown that deficiency of the glutamine transporter ASCT2 did not impact Tregs generation (**Nakaya et al., 2014**), which raises the question of the importance of Gln metabolism in Tregs.

Based on these observations, we tracked simultaneously Glc and Gln metabolism in WT and $Pgd^{fl/fl}Foxp3^{Cre}$ Tregs, both as direct isolates from mouse models with inflammatory disorder and also Tregs generated from naïve CD4$^+$ T cells in vitro. In the first set, YFP$^+$ Tregs were sorted from WT and $Pgd^{fl/fl}Foxp3^{Cre}$ mice and cultured in the presence of IL-2 (700 IU/ml) and anti-CD3/anti-CD28 coated beads (Treg:beads ratio 1:3) plus $D_7$-Glc and $^{13}C_5,^{15}N_2$-Gln for 48 hr. In the second set, naïve CD4$^+$ from WT and $Pgd^{fl/fl}Foxp3^{Cre}$ mice were polarized with IL-2+ TGF-β for 4 days and the medium was changed to that containing $D_7$-Glc + $^{13}C_5,^{15}N_2$-Gln for the last 48 hr. Transformations of $D_7$-Glc and $^{13}C_5,^{15}N_2$-Gln

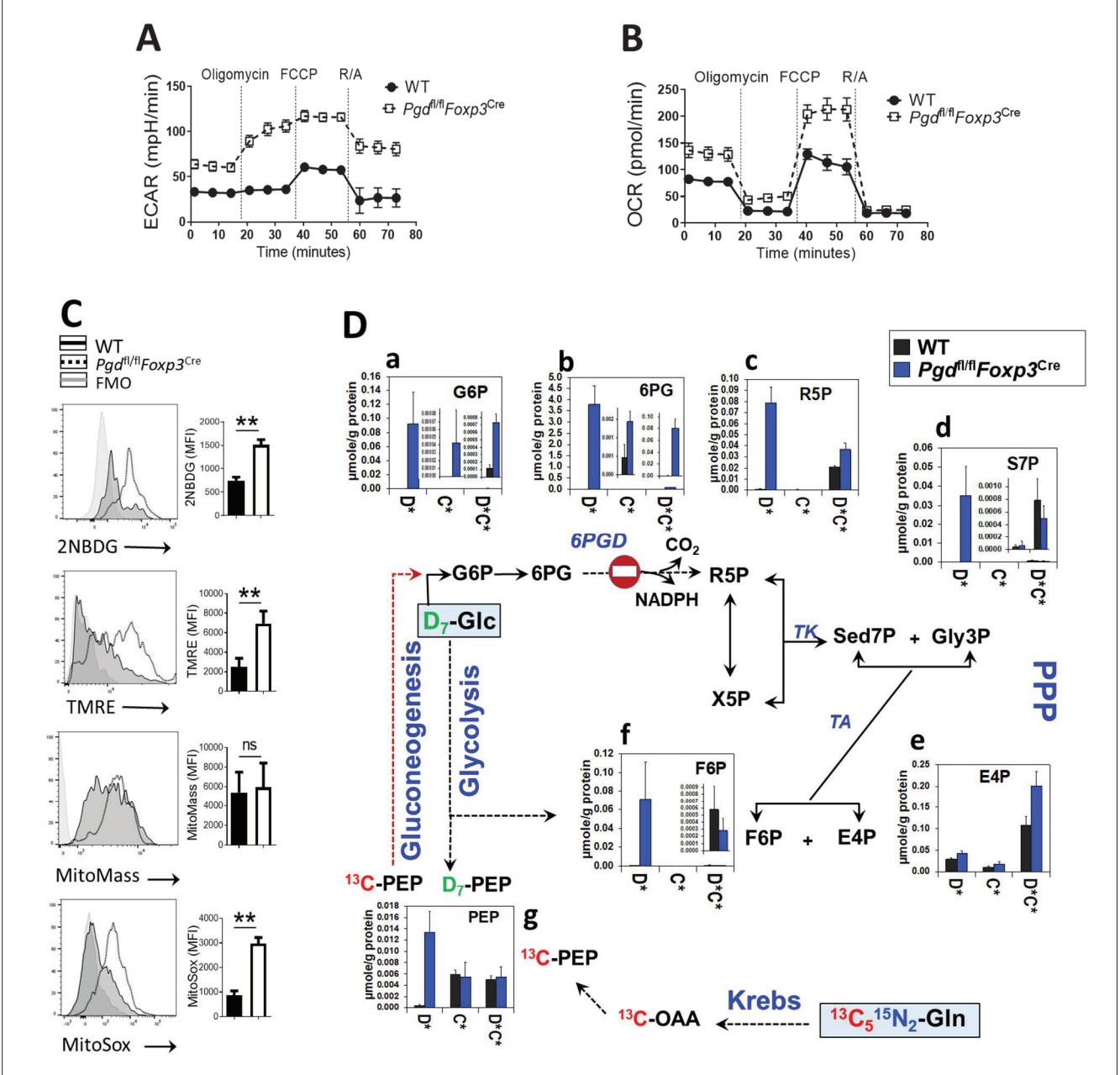

**Figure 4.** Blocking 6-phosphogluconate dehydrogenase (6PGD) in regulatory T cells (Tregs) induces reprogramming of glycolysis, mitochondrial respiration, and non-oxidative pentose phosphate pathway (PPP). (**A–B**) Extracellular acidification and oxygen consumption analysis on YFP+ sorted cells from $Pgd^{fl/fl}$ $Foxp3^{Cre}$ and wild-type (WT) mice was performed using Seahorse XF24 metabolic analyzer as described in supplemental information. The extracellular acidification rate (ECAR) (**A**) and oxygen consumption rate (OCR) (**B**) are shown (N = 4). (**C**) YFP+ cells glucose (Glc) uptake capacity was determined by 2-NBDG uptake, mitochondrial potential ($\Delta\Psi$m) by TMRE, mitochondrial mass by MitoTracker Deep Red FM and mitochondrial ROS by MitoSox Red. Results are representative of three independent experiments with N = 4 per group. (**D**) YFP+ cells were sorted from WT and $Pgd^{fl/fl}Foxp3^{Cre}$ mice and cultured in the presence of IL-2 (700 IU/ml) and anti-CD3/anti-CD28 coated beads (Treg:beads ratio 1:3) plus $D_7$-Glc and $^{13}C_5,^{15}N_2$-glutamine (Gln) for 48 hr. Isotope labeling patterns of metabolites of cell extracts were analyzed by IC-UHRMS. Data shown demonstrates $D_7$-Glc and $^{13}C_5,^{15}N_2$-Gln incorporation into the PPP metabolites via non-oxidative PPP and gluconeogenesis. Results were generated with N = 2. Legend in X-axis D* = sum of $D_1$ to $D_x$ or Glc-derived species; C* = sum of $^{13}C_1$ to $^{13}C_x$ or Gln-derived species; C*D* = sum of dual $^{13}C_1$ to $^{13}C_x$ and $D_1$ to $D_x$ or Glc and Gln-derived species; G6P, glucose-6-phosphate; 6 PG, 6-phosphogluconate; R5P, ribose-5-phosphate; S7P, sedoheptulose-7-phosphate; Gly3P, glyceraldehyde-3-phosphate; X5P, xylulose-5-phosphate; E4P, erythrose-4-phosphate; F6P, fructose-6-phosphate; OAA, oxaloacetate; PEP, phosphoenolpyruvate.

The online version of this article includes the following figure supplement(s) for figure 4:

**Figure supplement 1.** Glycolysis, Krebs cycle, and nucleotide biosynthesis are enhanced in 6-phosphogluconate dehydrogenase (6PGD)-deficient

*Figure 4 continued on next page*

*Figure 4 continued*

Tregs.

**Figure supplement 2.** Pentose phosphate pathway (PPP) pathway reprogramming in 6-phosphogluconate dehydrogenase (6PGD)-deficient regulatory T cells (Tregs) generated in vitro.

**Figure supplement 3.** Glycolysis, Krebs cycle, and nucleotide biosynthesis reprogramming in 6-phosphogluconate dehydrogenase (6PGD)-deficient Tregs generated in vitro.

in sorted Tregs into metabolites of PPP/gluconeogenesis (GNG) (*Figure 4D*), glycolysis (*Figure 4—figure supplement 1A*), Krebs cycle (*Figure 4—figure supplement 1B*), and the nucleotide synthesis pathway (*Figure 4—figure supplement 1C*) were determined by SIRM analysis. Shift in the same metabolic pathways examined in in vitro generated Tregs is presented in *Figure 4—figure supplement 2* and *Figure 4—figure supplement 3*.

SIRM analysis showed higher buildup of D-labeled glycolytic intermediates (*Figure 4—figure supplement 1A*), which together with enhanced ECAR (*Figure 4A*) pointed to increased glycolytic capacity in 6PGD blocked Tregs. These cells also displayed increased levels of D and/or $^{13}$C-labeled Krebs cycle metabolites (*Figure 4—figure supplement 1B*), which together with the observed increased OCR and mitochondrial potential (*Figure 4B–C*) indicated elevated Krebs cycle activity. Notably, Gln served as the main fuel source for the Krebs cycle (*Figure 4—figure supplement 1B*), with Gln metabolism being significantly higher in 6PGD-deficient than WT Tregs. Enhanced Gln metabolism via the Krebs cycle led to increased αKG production (*Figure 4—figure supplement 1B-F*), which could serve as critical 'metabolic signal' for CD4$^+$ T subset differentiation (*Klysz et al., 2015*). αKG level has been shown to control methylation of the *Foxp3* gene locus resulting in a shift of balance between Treg/Th1 (*Klysz et al., 2015*) and Treg/Th17 (*Xu et al., 2017*). Whether such epigenetic metabolic mechanism is involved in skewing 6PGD blocked Tregs toward Teff subsets awaits further investigation.

## 6PGD deficiency in Tregs enhances non-oxidative PPP and GNG

SIRM analysis also showed that 6PGD-deficient Tregs extensively accumulated G6P and 6 PG (*Figure 4D–a and b*), which confirmed effective blockade of oxidative PPP at 6PGD (*Figure 4—figure supplement 1A*). However, instead of depletion, this blockade led to enhanced accumulation of D and/or $^{13}$C-labeled R5P (*Figure 4D–c*), which are downstream products of the 6PGD reaction. Also D and/or $^{13}$C-labeled sedoheptulose-7-phosphate (S7P) (*Figure 4D–d*), erythrose-4-phosphate (*Figure 4D–e*), and fructose-6-phosphate (F6P) (*Figure 4D–f*) accumulated in 6PGD-deficient Tregs. These metabolites are all products of the non-oxidative PPP. These results point to enhanced non-oxidative PPP that results in elevated production of R5P from D$_7$-Glc and $^{13}$C$_5$,$^{15}$N$_2$-Gln (*Figure 1—figure supplement 1A*), despite the blockade at 6PGD. In turn, R5P generated fueled the enhanced biosynthesis of both pyrimidine and purine nucleotides (D, $^{13}$C, and $^{15}$N labeled species, *Figure 4—figure supplement 1C*) to fulfill the nucleotide demands of activated T cells.

It should be noted that enhanced incorporation of Gln derived carbon (C* or D*C*/C*NxDx scrambled species) into 6 PG (C* and D*C* scrambled species, *Figure 4D–b*) and ATP/GTP (C*, D*C*, and C*NxDx, *Figure 4—figure supplement 1C-m,l*) in 6PGD-deficient Tregs are consistent with elevated GNG in these cells. This is because these $^{13}$C-labeled species can only be produced via the reaction sequence of glutaminolysis, the Krebs cycle, and GNG initiated by conversion of OAA to PEP via PEPCK, followed by the reversal of the glycolytic reactions. The latter produce 3-phosphoglycerate, glyceraldehyde-3-phosphate, F6P, and G6P, which can fuel both oxidative and non-oxidative PPP as well as serine biosynthesis-one carbon pathway to power purine biosynthesis (*Figure 4—figure supplement 1A*). Increased accumulation of $^{13}$C-G6P, together with that of D-G6P (*Figure 4D–a*) indicate enhanced activity of G6PD, the first rate-limiting and NADPH-producing enzyme of the oxidative branch of the PPP. This is expected to help compensate for the loss of NADPH production due to blocked 6PGD action. Overall, similar shifts of metabolic patterns in Tregs generated in vitro (iTregs in *Figure 4—figure supplement 2* and *Figure 4—figure supplement 3*) as those sorted from mouse models were evident.

## Discussion

In our pilot studies, we found that inhibition of G6PD led to reduced T cell survival with little changes in IFN-γ production (*Figure 1—figure supplement 1B-C*) but that of 6PGD resulted in higher T cell survival and T subsets activations (*Figure 1—figure supplement 1B-C,I-J* ). Our G6PD inhibitor study is consistent with a recent study, which showed that G6PD inhibition depleted NADPH and decreased inflammatory cytokine production by T cells (*Ghergurovich et al., 2020*). To better understand the role of 6PGD in T cells, we generated *Pgd*fl/fl*Foxp3*Cre mice to examine the effect of 6GPD blockade on Tregs as central immune regulators. Interestingly, *Pgd*fl/fl*Foxp3*Cre mice showed severe autoimmune disorders highlighting a critical role of 6PGD in Tregs function (*Figure 1*). Tregs isolated from *Pgd*fl/fl*Foxp3*Cre mice showed reduced suppressive function in vitro and in vivo (*Figure 2*), which was accompanied by a shift of gene expression signatures and phenotype markers toward T CD4+ subsets of Th1, Th2, and Th17. Loss of suppressive function and plasticity toward conventional subsets was also reproduced by treatments with 6PGD small molecule inhibitor 6-AN (*Figure 3*).

Although Glc, but not Gln, contributes to Tregs differentiation and function (*Pacella et al., 2018*; *Procaccini et al., 2016*; *Nakaya et al., 2014*), Glc depletion by blocking the Glc transporter Glut1 did not alter the suppressive function of Tregs (*Michalek et al., 2011*; *Macintyre et al., 2014*). However, our study showed that the suppressive function of Tregs was modulated by altered Glc metabolism via the oxidative PPP, that is, *6PGD* deficiency in Tregs resulting in significantly improved anti-tumor responses (*Figure 3F–N*) or changed IBD outcomes (*Figure 2*). This qualifies 6PGD as an immune checkpoint by modulating the Treg function while implicating the importance of Glc metabolism via the oxidative PPP in Treg activation.

As our knowledge of Glc and Gln metabolism in Tregs is limited, we tracked the metabolic fates of both fuel substrates with stable isotope tracers in 6PGD blocked versus WT Tregs. We found that blocking 6PGD in oxidative PPP shifted Glc flow into glycolysis and non-oxidative PPP to provide sufficient substrates (e.g. one-carbon metabolites and R5P) for fueling nucleotide biosynthesis (*Figure 4* and *Figure 4—figure supplement 1*). This was also accompanied by enhanced Gln flow into GNG to power nucleobase synthesis and NADPH production, which can compensate for NADPH loss due to 6PGD blockade. These reprogrammed metabolic activities co-occurred with the shift of expression in phenotypic and functional markers toward conventional T CD4+ subsets. The importance of glycolytic shift on modulating Tregs fitness has been shown in previous studies (*Watson et al., 2021*; *Zappasodi et al., 2021*). Tregs in high Glc environment lose their fitness while upregulation of lactate metabolism is important for maintaining their suppressive phenotype (*Watson et al., 2021*). Also, the presence of Glc was shown to be a key factor in interfering Treg function under anti-CTLA-4 treatment (*Zappasodi et al., 2021*). Moreover, although glycolysis is key to supporting the energy and anabolic demands during Treg expansion (*Pacella et al., 2018*; *De Rosa et al., 2015*), further increase in glycolysis as a result of mTOR and c-Myc activation suppresses Treg generation (*Kabat et al., 2016*; *Wei et al., 2016*). Our metabolic fate analysis not only corroborates with glycolytic control of Treg functions but also reveals the role of 6PGD as a key regulator of Treg stability.

Although disruption of Gln uptake did not impact Treg function (*Nakaya et al., 2014*), enhanced Gln metabolism in 6PGD-deficient Tregs was evident and co-occurred with the shift from Treg to Th1 (*Michalek et al., 2011*; *Klysz et al., 2015*), Th2 (*Michalek et al., 2011*), or Th17 (*Michalek et al., 2011*; *Xu et al., 2017*) phenotypes. Increased Gln oxidation in the Krebs cycle promoted the synthesis of immunomodulatory metabolites such as αKG (*Figure 4—figure supplement 1*), which has been shown to signal Foxp3 suppression (*Klysz et al., 2015*; *Xu et al., 2017*). At the same time, diversion of Gln carbons for enhanced nucleotide synthesis would help meet the metabolic demand of the shift toward conventional T subsets. Thus, Gln metabolism could synergize with 6PGD deficiency in eliciting Treg plasticity toward immune activation. Further studies will be required to substantiate such metabolic mechanism.

In summary, we reported a critical role of 6PGD, the second rate-limiting enzyme of PPP, in skewing Tregs function and metabolism toward those of the Th1, Th2, and Th17 subsets. As such, 6PGD could be a novel metabolic target in clinical applications such as autoimmune disorders or cancer therapies.

# Materials and methods

## Key resources table

| Reagent type (species) or resource | Designation | Source or reference | Identifiers | Additional information |
|---|---|---|---|---|
| Antibody | Anti-mouse CD3 ε -PerCP (clone: 145–2 C11); (Armenian hamster monoclonal) | BioLegend | Cat No#100326; RRID: AB_893317 | FACS (1:100) |
| Antibody | Anti-mouse CD4-APC (clone: GK1.5); (rat monoclonal) | BioLegend | Cat No#100412; RRID: AB_312697 | FACS (1:100) |
| Antibody | Anti-mouse CD4-PE (clone: GK1.5); (rat monoclonal) | BioLegend | Cat No#100408; RRID: AB_312693 | FACS (1:100) |
| Antibody | Anti-mouse CD4-Pacific Blue (clone: GK1.5); (rat monoclonal) | BioLegend | Cat No#100428; RRID: AB_ 493,647 | FACS (1:100) |
| Antibody | Anti-mouse CD8a-FITC (clone: 53–6.7); (rat monoclonal) | BioLegend | Cat No#100706; RRID: AB_312745 | FACS (1:100) |
| Antibody | Anti-mouse CD8a-Pacific Blue(clone: 53–6.7); (rat monoclonal) | BioLegend | Cat No#100725; RRID: AB_493425 | FACS (1:100) |
| Antibody | Anti-mouse CD25-BV711 (clone: PC61); (rat monoclonal) | BioLegend | Cat No#102049; RRID: AB_2564130 | FACS (1:100) |
| Antibody | Anti-mouse CD11b-Pacific Blue (clone: M1/70); (rat monoclonal) | BioLegend | Cat No#101224; RRID: AB_755986 | FACS (1:100) |
| Antibody | Anti-mouse CD11c-PE-Cy7 (clone: N418); (Armenian hamster monoclonal) | BioLegend | Cat No#117318; RRID: AB_493568 | FACS (1:100) |
| Antibody | Anti-mouse CD25-PE-Cy7 (clone: PC61); (rat monoclonal) | BioLegend | Cat No#102016; RRID: AB_312865 | FACS (1:100) |
| Antibody | Anti-mouse CD40L-APC (clone: MR1); (Armenian hamster monoclonal) | BioLegend | Cat No#106510; RRID: AB_2561561 | FACS (1:100) |
| Antibody | Anti-mouse CD44-PE-Cy7 (clone: IM7); (rat monoclonal) | BioLegend | Cat No#103030; RRID: AB_830787 | FACS (1:100) |
| Antibody | Anti-mouse CD45.2-PE-Cy7 (clone: 104); (mouse monoclonal) | BioLegend | Cat No#109830; RRID: AB_1186098 | FACS (1:100) |
| Antibody | Anti-mouse CD71-APC (clone: RI7217); (rat monoclonal) | BioLegend | Cat No#113820; RRID: AB_2728135 | FACS (1:100) |
| Antibody | Anti-mouse CD98-PE-Cy7 (clone: RL388); (rat monoclonal) | BioLegend | Cat No#128214; RRID: AB_2750547 | FACS (1:100) |
| Antibody | Anti-mouse CD45.2-PerCP (clone: 104); (mouse monoclonal) | BioLegend | Cat No#109826; RRID: AB_893349 | FACS (1:100) |
| Antibody | Anti-mouse CD62L-Pacific Blue (clone: MEL-14); (rat monoclonal) | BioLegend | Cat No#104424; RRID: AB_493380 | FACS (1:100) |
| Antibody | Anti-mouse CD69-PE (clone: H1.2F3); (Armenian hamster monoclonal) | BioLegend | Cat No#104508; RRID: AB_313111 | FACS (1:100) |
| Antibody | Anti-mouse CD107a-APC-Cy7 (clone: 1D4B); (rat monoclonal) | BioLegend | Cat No#121616; RRID: AB_10643268 | FACS (1:100) |
| Antibody | Anti-mouse Foxp3-PE (clone: MF-14); (rat monoclonal) | BioLegend | Cat No#126404; RRID: AB_1089117 | FACS (1:100) |
| Antibody | Anti-mouse siglec-F (CD170)-APC (clone: S17007L); (rat monoclonal) | BioLegend | Cat No#155508; RRID: AB_2750237 | FACS (1:100) |
| Antibody | Anti-mouse Ly6C-APC (clone: HK1.4); (rat monoclonal) | BioLegend | Cat No#128016; RRID: AB_1732076 | FACS (1:100) |
| Antibody | Anti-mouse Ly6G-PE (clone: 1A8); (rat monoclonal) | BioLegend | Cat No#127608; RRID: AB_1186099 | FACS (1:100) |
| Antibody | Anti-mouse IL-4-PE (clone: 11B11); (rat monoclonal) | BioLegend | Cat No#504104; RRID: AB_315318 | FACS (1:100) |
| Antibody | Anti-mouse IL-5-APC (clone: TRFK5); (rat monoclonal) | BioLegend | Cat No#504306; RRID: AB_315330 | FACS (1:100) |

*Continued on next page*

*Continued*

| Reagent type (species) or resource | Designation | Source or reference | Identifiers | Additional information |
|---|---|---|---|---|
| Antibody | Anti-mouse IL-13-PE (clone: eBio13A); (rat monoclonal) | ThermoFisher | Cat No# 12-7133-82; RRID: AB_763559 | FACS (1:100) |
| Antibody | Anti-mouse IL-17A-PE (clone: eBio17B7); (rat monoclonal) | ThermoFisher | Cat No#12-7177-81; RRID: AB_763582 | FACS (1:100) |
| Antibody | Anti-mouse IFN-γ-APC (clone: XMG1.2); (rat monoclonal) | BioLegend | Cat No#505810; RRID: AB_315404 | FACS (1:100) |
| Antibody | Anti-mouse Granzyme B-PE (clone: NGZB); (rat monoclonal) | ThermoFisher | Cat No#12-8898-82; RRID: AB_10870787 | FACS (1:100) |
| Antibody | Anti-mouse CD3 ε -Purified (clone: 145–2 C11); (Armenian hamster monoclonal) | BioLegend | Cat No#100340; RRID: AB_11149115 | Culture (1:1000) |
| Antibody | Anti-mouse CD28-Purified (clone: 37.51); (Syrian hamster monoclonal) | BioLegend | Cat No#102116; RRID: AB_11147170 | Culture (1:1000) |
| Antibody | Anti-mouse 6PGD; (rabbit polyclonal) | Sigma Aldrich | Cat No#HPA031314; RRID: AB_10610278 | Western blot (1:1000) |
| Antibody | Anti-β-actin (D6A8); (rabbit monoclonal) | Cell Signaling Technology | Cat No#8457; RRID: AB_10950489 | Western blot (1:1000) |
| Genetic reagent (species) | *Pgd* TaqMan Assay probe (FAM-MGB) (mouse) | ThermoFisher | Cat No#4351372; Assay ID: Mm01263703_m1 | |
| Genetic reagent (species) | *Foxp3* TaqMan Assay probe (FAM-MGB) (mouse) | ThermoFisher | Cat No#4331182; Assay ID: Mm00475162_m1 | |
| Genetic reagent (species) | Gata3 TaqMan Assay probe (FAM-MGB) (mouse) | ThermoFisher | Cat No#4331182; Assay ID: Mm00484683_m1 | |
| Genetic reagent (species) | *Il5* TaqMan Assay probe (FAM-MGB) (mouse) | ThermoFisher | Cat No#4331182; Assay ID: Mm00439646_m1 | |
| Genetic reagent (species) | *Il13* TaqMan Assay probe (FAM-MGB) (mouse) | ThermoFisher | Cat No#4331182; Assay ID: Mm00434204_m1 | |
| Genetic reagent (species) | *Ifng* TaqMan Assay probe (FAM-MGB) (mouse) | ThermoFisher | Cat No#4331182; Assay ID: Mm01168134_m1 | |
| Genetic reagent (species) | *Tbet* (*Tbx21*) TaqMan Assay probe (FAM-MGB) (mouse) | ThermoFisher | Cat No# 4331182; Assay ID: Mm00450960_m1 | |
| Genetic reagent (species) | *Il17a* TaqMan Assay probe (FAM-MGB) (mouse) | ThermoFisher | Cat No#4331182; Assay ID: Mm00439618_m1 | |
| Genetic reagent (species) | *Icos* TaqMan Assay probe (FAM-MGB) | ThermoFisher | Cat No#4331182; Assay ID: Mm00497600_m1 | |
| Genetic reagent (species) | *Il4* TaqMan Assay probe (FAM-MGB) (mouse) | ThermoFisher | Cat No#4331182; Assay ID: Mm00445259_m1 | |
| Genetic reagent | *Il10* TaqMan Assay probe (FAM-MGB) (mouse) | ThermoFisher | Cat No#4331182; Assay ID: Mm00439614_m1 | |
| Genetic reagent (species) | *Batf3* TaqMan Assay probe (FAM-MGB) (mouse) | ThermoFisher | Cat No#4331182; Assay ID: Mm01318274_m1 | |
| Genetic reagent (species) | *Hey1* TaqMan Assay probe (FAM-MGB) (mouse) | ThermoFisher | Cat No#4331182; Assay ID: Mm00468865_m1 | |
| Genetic reagent (species) | *Bcl6* TaqMan Assay probe (FAM-MGB) (mouse) | ThermoFisher | Cat No#4331182; Assay ID: Mm00477633_m1 | |
| Genetic reagent (species) | *Gzb* TaqMan Assay probe (FAM-MGB) (mouse) | ThermoFisher | Cat No#4331182; Assay ID: Mm00442834_m1 | |
| Genetic reagent (species) | 18 S rRNA TaqMan Assay probe (VIC-MGB) (mouse) | ThermoFisher | Cat No#4319413E | |
| Cell line (species) | B16-F10 (mouse) | ATCC | ATCC CRL-6475; RRID:CVCL_0159 | |

*Continued on next page*

*Continued*

| Reagent type (species) or resource | Designation | Source or reference | Identifiers | Additional information |
|---|---|---|---|---|
| Chemical compound, drug | 6-Aminonicotinamide (6-AN) | Sigma Aldrich | Cat No#A68203 | |
| Chemical compound, drug | Tamoxifen | Sigma Aldrich | Cat No#T5648-1G | |
| Chemical compound, drug | Dehydroepiandrosterone (DHEA) | Cayman Chemical | Cat No#15728 | |
| Chemical compound, drug | Dimethyl sulfoxide (DMSO) | Sigma Aldrich | Cat No#D2438 | |
| Chemical compound, drug | Tetra-methylrhodamine ester (TMRE) | ThermoFisher | Cat No#T669 | |
| Chemical compound, drug | MitoSOX Red | ThermoFisher | Cat No#M36008 | |
| Chemical compound, drug | MitoTracker Deep Red FM | ThermoFisher | Cat No#M22426 | |
| Chemical compound, drug | 2-NBD-glucose (2-NBDG) | Cayman Chemical | Cat No#11046 | |
| Chemical compound, drug | D-GLUCOSE (1,2,3,4,5,6,6-D7, 97–98%) | Cambridge Isotope laboratories | Cat No#DLM-2062-PK | |
| Chemical compound, drug | $^{13}C_5,^{15}N_2$-Glutamine | Cambridge Isotope laboratories | Cat No#CNLM-1275-H-PK | |
| Chemical compound, drug | Fetal bovine serum, heat inactivated | ThermoFisher | Cat No#16140071 | |
| Chemical compound, drug | NuPAGE 4–12%, Bis-Tris, 1.5 mm, Mini Protein Gel | ThermoFisher | Cat No#NP0335BOX | |
| Chemical compound, drug | NuPAGE MES SDS Running Buffer | ThermoFisher | Cat No#NP0002 | |
| Chemical compound, drug | Recombinant Mouse IL-2 | BioLegend | Cat No#575404 | |
| Chemical compound, drug | Recombinant Mouse TGF-β1 | BioLegend | Cat No#763104 | |
| Commercial assay or kit | Milliplex MAP Mouse Cytokine/Chemokine Magnetic Bead Panel – Premixed 32 Plex | Millipore | Cat No# MCYTMAG-70K-PX32 | |
| Commercial assay or kit | Mouse Ig Isotyping Array Q1 | RayBiotech | Cat No#QAM-ISO-1–2 | |
| Commercial assay or kit | CellTrace CFSE Cell Proliferation Kit | ThermoFisher | Cat No#C34554 | |
| Commercial assay or kit | Dynabeads Mouse T-Activator CD3/CD28 | ThermoFisher | Cat No#11456D | |
| Commercial assay or kit | Fixation/Permeabilization Solution Kit with BD GolgiPlug | BD Bioscience | Cat No#555028 | |
| Commercial assay or kit | eBioscience Foxp3/ Transcription Factor Staining Buffer Set | ThermoFisher | Cat No#00-5523-00 | |
| Commercial assay or kit | LIVE/DEAD Fixable Aqua Dead Cell Stain Kit (Aqua) | ThermoFisher | Cat No#L34957 | |
| Commercial assay or kit | EasySep Mouse CD4+ T Cell Isolation Kit | STEMCELL Technologies | Cat No#19852RF | |
| Commercial assay or kit | EasySep Mouse Naïve CD4+ T Cell Isolation Kit | STEMCELL Technologies | Cat No#19765 | |
| Commercial assay or kit | EasySep Mouse CD4+ CD25+ Regulatory T Cell Isolation Kit II | STEMCELL Technologies | Cat No#18783 | |
| Commercial assay or kit | RNeasy Mini Kit | Qiagen | Cat No#74104 | |
| Commercial assay or kit | Seahorse XF Cell Mito Stress Test Kit | Agilent | Cat No#103015–100 | |
| Commercial assay or kit | Pierce BCA Protein Assay Kit | ThermoFisher | Cat No#23225 | |

*Continued on next page*

*Continued*

| Reagent type (species) or resource | Designation | Source or reference | Identifiers | Additional information |
|---|---|---|---|---|
| Genetic reagent (*Mus musculus*) | *Pgd*fl/fl *Foxp3*YFP-Cre | This paper | N/A | *6PGD* exon 5 floxed. Send reagent request to pseth@bidmc.harvard.edu |
| Genetic reagent (*Mus musculus*) | C57BL/6 J (B6 CD45.2+) | The Jackson Laboratory | Stock No: 000664 | |
| Genetic reagent (*Mus musculus*) | B6.SJL-Ptprca Pepcb/BoyJ (B6 CD45.1+) | The Jackson Laboratory | Stock No: 002014 \| B6 Cd45.1 | |
| Genetic reagent (*Mus musculus*) | B6.129S7-Rag1tm1Mom/J | The Jackson Laboratory | Stock No: 002216 | |
| Genetic reagent (*Mus musculus*) | B6.129(Cg)-*Foxp3*tm4(YFP/icre)Ayr/J | The Jackson Laboratory | Stock No: 016959 | |
| Genetic reagent (*Mus musculus*) | *Foxp3*tm9(EGFP/cre/ERT2)Ayr/J | The Jackson Laboratory | Stock No: 016961 | |
| Software, algorithm | FlowJo_V10 | FlowJo | https://www.flowjo.com/; RRID:SCR_008520 | |
| Software, algorithm | GraphPad Prism_V6 | GraphPad | https://www.graphpad.com/; RRID:SCR_002798 | |

## Cell line

B16-F10 cell line was obtained from ATCC (ATCC CRL-6475). The cell line was tested for *Mycoplasma* contamination.

## Mice

Steps to generate of $Pgd^{fl/fl}Cd4^{Cre}$, $Pgd^{fl/fl}Foxp3^{Cre}$, and $Pgd^{fl/fl}$ $Foxp3^{EGFP\text{-}Cre\text{-}ERT2}$ mice are described in *Figure 1—figure supplement 1*. Verification of 6PGD deletion was done by PCR, western blot, and alteration in metabolite profile associated with the enzyme. Other mice used during experiments including C57BL/6 J (B6 CD45.2+), B6.SJL-Ptprca Pepcb/BoyJ (B6 CD45.1+), B6.129(Cg)-$Foxp3^{tm4(YFP/icre)Ayr}$/J ($Foxp3^{YFP\text{-}cre}$), $Foxp3^{tm9(EGFP/cre/ERT2)Ayr}$/J ($Foxp3^{eGFP\text{-}Cre\text{-}ERT2}$), and B6.129S7-$Rag1^{tm1Mom}$/J ($Rag1^{-/-}$) mice were purchased from Jackson Laboratories. All mice were kept in specific pathogen-free conditions prior to use. Animal work was done in accordance with the Institutional Animal Care and Use Committee of the University of BIDMC (Protocol number 040–2016).

## Detection of cells markers by flow cytometry

Flow cytometry stainings were done on prepared cells. Cells from spleen and lymph nodes were obtained by tissue harvest, mechanically disintegration, passing through 70 μm cell strainer and then red blood cell lysis. Tumor infiltration cells were also prepared using the same steps with additional 30 min collagenase treatment at 37 °C. Cultured cells and sorted cells just passed one step of wash with FACS buffer (PBS containing 1 % FBS). Cells were suspended in FACS buffer and stained for surface markers for 30 min at 4 °C. Aqua LIVE/DEAD Fixable Dead Cell Stain was used in all staining to distinguish dead cells. Intracellular staining for IL-4, IL-13, Il-5, IFN-γ, IL-17A, TGF-β, and granzyme B was performed with the Fixation/Permeabilization Solution Kit with BD GolgiPlug (BD Bioscience) according to the manufacturer's protocol. To detect cytokines, cells were cultured in RMPI containing 10 % FBS and were stimulated with PMA (50 ng/ml), ionomycin (1 μg/ml), and GolgiPlug (1 μl/ml) for 4 hr at 37 C before starting the staining. For assessment of Foxp3, cells were stained intracellularly using the Foxp3/Transcription Factor Buffer Set (eBiosciences) following the manufacturer's protocol. Glc uptake was analyzed by incubation of the cells with 20 μM 2-NBD-Glc (Cayman Chemical) for 30 min at 37 °C. Cells were analyzed by a CytoFLEX LX (Beckman Coulter) cytometer and analysis was performed using FlowJo_V10 software. All antibodies are listed in the Key resources table.

## Cell isolation by flow cytometry cell sorting

To obtain purified cells, single cells were first enriched by negative selection using EasySep Mouse CD4+ T cells isolation kit (STEMCELL Technologies) following manufacturer's protocols. Then cells were stained for desired surface markers and sorted by flow cytometry. Surface markers for gating/

sorting of each cell type included: naïve CD4$^+$ T cells (CD4$^+$ CD62L$^{high}$ CD44$^{low}$), YFP$^+$ Tregs (CD4$^+$ CD25$^+$ YFP$^+$), EFGP$^+$ Tregs (CD4$^+$ CD25$^+$ EGFP$^+$), and T effector cells (CD4$^+$ CD45RB$^{high}$). All the stainings were done at 4 °C for 30 min and sorting was done by Becton Dickinson SORP FACSAria II (Beckman Coulter). Isolated cell phenotypes were confirmed by running again with flow cytometer (CytoFLEX LX, Beckman Coulter).

## Cell culture and treatment

### Naïve CD4$^+$ T cell treatment with inhibitors

Naïve CD4$^+$ T cells (CD4$^+$ CD62L$^{high}$ CD44$^{low}$) were isolated and cultured in complete RPMI media containing 10 % FBS and IL-2 (20 IU/ml) and treated with 6-AN (10 µM), DHEA (10 µM), or vehicle DMSO. Cells stimulation was done by adding soluble anti-CD3 (1 µg/ml) and soluble anti-CD28 (1 µg/ml). Cells were harvested and evaluated 4 days post culture.

### Naïve CD4$^+$ T cells polarization toward Tregs and treatment with 6-AN

To target 6PGD in vitro, naïve CD4$^+$ T cells (CD4$^+$ CD62L$^{high}$ CD44$^{low}$) were isolated and polarized to Tregs using IL-2 (50 IU/ml) and TGF-β1 (5 ng/ml). Stimulation was done by culturing the cells in plate coated antibodies. For coating anti-CD3 and anti-CD28 (2 µg/ml PBS each) were put on the plate and incubated 4 hr at 37 °C and then washed twice with PBS. For inhibitor treatment 6-AN (10 µM) or vehicle DMSO was added.

### Culture of YFP$^+$ sorted cells

Isolated Tregs (CD4$^+$ CD25$^+$ YFP$^+$) were cultured in vitro in presence of IL-2 (700 IU/ml) and anti-CD3/anti-CD28 coated beads (Treg:beads ratio 1:3) and cells number was assessed at 24 and 48 hr time points. Media was evaluated for cytokines production.

## Suppression assay

Suppression assay was done on prepared Tregs as YFP$^+$ sorted cells from *6PGD*$^{+/+}$*Foxp3*$^{Cre}$ (WT) and *Pgd*$^{fl/fl}$*Foxp3*$^{Cre}$ mice, EGFP$^+$ sorted cells from *Pgd*$^{+/+}$ *Foxp3*$^{EGFP-Cre-ERT2}$ (WT) and *Pgd*$^{fl/fl}$ *Foxp3*$^{EGFP-Cre-ERT2}$ mice, and also DMSO (control) and 6-AN derived Tregs. These cells express CD45.2 on their surface (B6 CD45.2$^+$). To check suppressive capacity of prepared Tregs, the cells were mixed with WT CD45.1$^+$ effector T cells (CD4$^+$ CD45RB$^{high}$) in different ratio and cultured in U shaped plate. The effecter cells were stained with CFSE (0.5 mM) for 30 min before culture. For optimized stimulation of effector cells, $2.5 \times 10^5$ irradiated (30 Gy) splenocytes from CD45.2$^+$ mice (WT B6 CD45.2$^+$) and soluble anti-CD3 (1 µg/ml) was added. After 72 hr of co-culture, amount of CFSE dilution was determined in CD4$^+$ CD45.1$^+$ effector cells using flow cytometer (CytoFLEX LX, Beckman Coulter).

## RNA purification, RNAseq, and real-time quantitative PCR

Total RNA was extracted from YFP$^+$ sorted cells with RNeasy Mini Kit (Qiagen) and RNA concentrations were determined using Nanodrop (Thermo Scientific). Total DNA-free RNA was used for mRNA isolation and library construction. Libraries were sequenced on an Illumina HISEQ 2500 (Illumina). Confirmation of RNAseq was done by examination of key representative genes by real-time quantitative PCR method using ABI 7300 Real-Time PCR system (Applied Biosystems). Used TaqMan probes are listed in the Key resources table. Expression of each gene was normalized to the housekeeping gene expression (18 S rRNA) as $\Delta CT = CT$ (gene X) – CT (18 S rRNA). The alteration between groups was calculated as fold change = $2^{\wedge}(\Delta\Delta CT)$. $\Delta\Delta CT$ is: CT (gene X) – CT (gene X at baseline).

## Bioenergetics analysis by Seahorse and mitochondrial activity

Prepared Tregs (YFP$^+$ sorted cells) were seeded on Cell-Tak coated Seahorse XFe24 (Agilent) culture plates ($0.8–1 \times 10^6$ cells/well) in assay media. The media was DMEM supplemented with 1 % BSA and 25 mM Glc, 1 mM pyruvate, and 2 mM Gln. The ECAR and OCR were measured over 75 min. To obtain maximal respiratory and control values, cells were treated with oligomycin (1 µM), FCCP (1.5 µM), and rotenone/antimycin A (0.5 µM).

Mitochondrial activity also was examined by flow cytometry. For this, cells were stained with 200 nM MitoTracker Deep Red FM (ThermoFisher) for mitochondrial mass and 200 nM tetra-methylrhodamine ester (TMRE) (ThermoFisher) for mitochondrial membrane potential. Staining was done in FACS

buffer (PBS containing 2% FBS) along with cell surface markers for 30 min at 37 °C. To detect mitochondrial reactive oxygen species cells surface markers were added and then 5 µM MitoSOX Red (ThermoFisher) was added in last 10 min incubation at 37 °C. Cells were analyzed by a CytoFLEX LX (Beckman Coulter) cytometer and analyzed by FlowJo_V10.

## IBD model

To examine Tregs suppressive activities in vivo, the IBD model was used as described previously (*Ostanin et al., 2009*; *Steinbach et al., 2015*). In brief, YFP$^+$ cells from WT and *Pgd$^{fl/fl}$ Foxp3$^{Cre}$* mice and also CD4$^+$ effector T cells (CD4$^+$CD45RB$^{high}$) from WT mice were sorted and mixed. Cells were resuspended to $4 \times 10^6$ cells/ml effector (CD45$^{high}$) and $2 \times 10^6$ cells/ml Treg (YFP$^+$). Total of $4 \times 10^5$ CD45$^{high}$ and $2 \times 10^5$ Treg was injected into each *Rag1$^{-/-}$* (B6.129S7-*Rag1$^{tm1Mom}$*/J) mice intra-peritoneally (i.p.). Control positive group was injection of effector cells alone and negative control group was *Rag1$^{-/-}$* mice without injection. Disease progress was monitored by weight loss, change in stools form, and diarrhea. Colons length and thickness along with colon's histochemistry were evaluated 45 days post cells injection.

## SIRM analysis on Tregs

To trace Glc and Gln in Tregs, YFP$^+$ sorted cells from WT and *Pgd$^{fl/fl}$Foxp3$^{Cre}$* mice were cultured in presence of IL-2 (700 IU/ml) and anti-CD3/anti-CD28 coated beads (Treg:beads ratio 1:3) and D$_7$-Glc plus $^{13}C_5,^{15}N_2$-Gln for 48 hr. To prepare labeling media, Glc-free Gln-free RPMI medium was supplemented with 10 % dialyzed FBS (Life Technologies), 20 mM HEPES, 0.05 mM 2-mercaptoethanol, and 1 % penicillin-streptomycin. Isotope labeling patterns of metabolites of cell extracts were analyzed by Ion Chromatography-UltraHigh Resolution Mass Spectrometry. The procedures were followed as previously described (*Fan et al., 2016b*; *Fan et al., 2016a*).

## Statistical analysis

All the statistical analyses were done using GraphPad Prism 6 (GraphPad) and Excel software. Difference between groups was analyzed using the Student's t test (for two groups) or one-way ANOVA (for multiple groups), followed by the post hoc Tukey test. Differences in tumor growth was calculated by two-way ANOVA. Differences between groups were rated significant at values of $p < 0.05$. In the figures all data are shown as mean ± SEM and *$p \leq 0.05$, **$p \leq 0.01$, ***$p \leq 0.001$.

## Acknowledgements

This work was supported by BIDMC seed funds to PS, 2014-07-1112 target grant from Bayer to PS, and a pilot grant to PS from 1U24DK097215-01A1 (to RMH and TWMF), and the Markey Cancer Center Redox and Metabolism Shared Resource Facility P30CA177558.

## Additional information

### Funding

| Funder | Grant reference number | Author |
| --- | --- | --- |
| Bayer Fund | 2014-07-1112 | Pankaj Seth |
| NIH Office of the Director | 1U24DK097215-01A1 | Teresa Fan |
| National Cancer Institute | P30CA177558 | Teresa Fan |

The funders had no role in study design, data collection and interpretation, or the decision to submit the work for publication.

### Author contributions

Saeed Daneshmandi, Conceptualization, Data curation, Experimental design and execution, Mouse models and immunological designs and analysis, Metabolic pathway interpretation, Formal analysis, Investigation, Methodology, Project administration, Visualization, Writing – original draft, Writing

– review and editing; Teresa Cassel, Data curation, Investigation, SIRM analyses; Richard M Higashi, Data curation, Investigation, SIRM design and analyses; Teresa W-M Fan, Conceptualization, Contributed to SIRM experimental design and execution, Metabolic pathway interpretation, Data curation, Investigation, Methodology, Writing – review and editing; Pankaj Seth, Conceptualization, Data curation, Funding acquisition, Investigation, Methodology, Resources, Supervision, Writing – original draft, Writing – review and editing

### Author ORCIDs

Saeed Daneshmandi (ID) http://orcid.org/0000-0001-7817-3006
Teresa Cassel (ID) http://orcid.org/0000-0003-1700-932X
Teresa W-M Fan (ID) http://orcid.org/0000-0002-7292-8938

### Ethics

Animal work was done in accordance with the Institutional Animal Care and Use Committee of Beth Israel Deaconess Medical Center.(Protocol number 040-2016).

### Decision letter and Author response

Decision letter https://doi.org/10.7554/eLife.67476.sa1
Author response https://doi.org/10.7554/eLife.67476.sa2

## Additional files

### Supplementary files

• Transparent reporting form

### Data availability

RNAseq data was deposited to NCBI BioSample (BioSample Accession # PRJNA706901).

The following dataset was generated:

| Author(s) | Year | Dataset title | Dataset URL | Database and Identifier |
|---|---|---|---|---|
| Daneshmandi S, Cassel T, Higashi RH, Fan TWM, Seth P | 2021 | 6-Phosphogluconate dehydrogenase (6PGD) a key checkpoint in reprogramming of Treg metabolism and function | https://www.ncbi.nlm.nih.gov/sra/PRJNA706901 | NCBI Sequence Read Archive, PRJNA706901 |

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
