## [Decision Letter]

**Acceptance summary:**

The study by Daneshmandi et al, advances knowledge of immune-metabolism, describing a key role for the pentose phosphate pathway in the expression of the transcription factor Foxp3, and functional consequences in a cell type that regulates prolonged inflammation in the mammalian immune system. Using a combination of in vitro and in vivo analysis and multiple approaches, included the detailed analysis of metabolic pathways, the study identifies a metabolic checkpoint in immune cell differentiation and function. The study will be of interest to immunologists, cell biologists and those with interests in metabolic networks underpinning cellular differentiation and function.

**Decision letter after peer review:**

Thank you for submitting your article "6-Phosphogluconate dehydrogenase (6PGD) a key checkpoint in reprogramming of Treg metabolism and function" for consideration by *eLife*. Your article has been reviewed by 3 peer reviewers, including Apurva Sarin as the Reviewing Editor and Reviewer #1, and the evaluation has been overseen by Satyajit Rath as the Senior Editor. The following individual involved in review of your submission has agreed to reveal their identity: Silvia Piconese (Reviewer #2).

Summary:

This study demonstrates that 6PGD, a rate-limiting enzyme in the oxidative pentose-phosphate pathway, modulates Treg metabolism and functions. Mice with a Treg-restricted 6PGD deficiency (either constitutive or inducible) spontaneously develop a multi-organ scurfy-like lethal disease, and succumb to early death, suggesting a loss of Treg cell immunosuppressive function. Treg cell frequency in these mice is apparently only slightly reduced and is mostly affected by a transcriptional reprogramming to resemble that of conventional T effector subsets, with Tregs presenting skewed metabolism toward glycolysis, Krebs cycle, and non-oxidative PPP. The study includes a description of disease phenotypes and cellular properties, which is supported by transcriptional analysis and detailed metabolite analysis at the cellular level. The precise mechanism through which 6PGD destabilizes Tregs remains to be elucidated. The core ideas of this manuscript are of interest in the area of Treg immunometabolism, which currently lags behind conventional CD4 and CD8 literature.

Daneshmandi et al. describe metabolic and functional consequences of blocking the oxidative pentose phosphate pathway (PPP) in Treg cells. While the core ideas of this manuscript are of interest in the area of Treg immunometabolism, there are substantial issues – outlined in the comments, that remain to be addressed.

Essential revisions:

1. Most experiments describing the Treg cell phenotype and function are in Foxp3Cre 6PGDfl/fl mice, which has significant systemic inflammation. This confounds Treg cell phenotypes that are presented (e.g., RNA-seq profile, in vitro suppression, ECAR, OCR, adoptive transfer). The Foxp3Cre 6PGDfl/fl system is appropriate for the in vivo profiling data of conventional CD4, CD8, and Treg cells, but not to conclude direct effects of 6PGD loss within Treg cells. The characterization of 6PGD deficient Treg cells should be undertaken in an inflammation-free system, using either mixed bone marrow chimera mice or an inducible-cre system (see the following comment). Alternatively, key analyses may be performed using Foxp3Cre/+ heterozygous females, only if they do not develop inflammation. Characterization of the Treg phenotype in an inflammation-free background is critical to support the main conclusion of the study.

2. The authors are advised to reassess the experiments using tamoxifen-inducible Foxp3-CreERT2 as the intricacies of this system appear to have been overlooked. Tamoxifen treatment does not induce Foxp3-EGFP expression in these mice as Foxp3-EGFP is expressed irrespective of tamoxifen (doi: 10.1126/science.1191996). Tamoxifen treatment will induce Cre-mediated gene deletion in Foxp3 expressing cells, but the cells are already EGFP+. In order to observe these cells, the mice must be crossed with a Rosa26-driven, stop-floxed fluorescent reporter mouse (e.g., YFP, available at jax.org/strain/006148), which will have fluorescent reporter expression only in the cells with successful Cre activation. The bulk EGFP+ cells contain a mixture of 6PGD+/+ and 6PGD-/- Treg cells. Furthermore, Foxp3-CreERT2 is a notoriously inefficient system for successful Cre activation, with only 50% or less of Treg cells actually having Cre-mediated gene deletion following tamoxifen treatment in vivo. If there are constraints on completing the experiment with the reporter as recommended, the data in its current form may be omitted from the study.

3. In the Results section, the sentence "As our preliminary studies showed that 6PGD but not G6PD blockade induces T cells activation (Figure 1—figure supplement 1 B-C). " is truncated. The preliminary data shown in suppl. Figure 1B-C indicates that a 6PGD pharmacological inhibitor enhances IFN-γ production by in vitro activated CD4 T cells, not that it "induces" T cell activation: the text should be corrected accordingly. In supplemental figure 1, the effects of DHEA on cell viability are noticeable. Any activation defects on the cells are likely a secondary consequence. Is it possible to use a lower dose to avoid so much cell death? In Figure 1G, the sentence "Lymphoproliferation was attributed to a higher number of CD4^+^ and CD8^+^ T cells in spleen/lymphoid organs should be corrected, since this figure shows frequencies and not counts. Further, Figure 1G only shows one plot, but it is from multiple organs according to the text. Are the other plots missing?

4. Regarding Treg frequency in Figure 2A, it is more informative to show the percentage of Foxp3+ in gated CD4 T cells, and also to show the gMFI of Foxp3 in gated Foxp3+ cells (it seems much lower in fl/fl mice; indeed this is confirmed by RT-PCR in Figure 2E). Is the difference between the two groups statistically significant? Statistics for frequencies and total numbers of Treg cells are missing. It is unclear why iTreg cells were used for figure 2C? A 50% reduction in Foxp3 gene expression, (Figure 2E), is substantial and seems incongruent with the claim of only a "slight decrease" in YFP MFI. Does direct Foxp3 staining and MFI quantification show the same result?

5. Figure 2—figure supplement 1D does not explain Treg plasticity since it refers to conventional effector T cell polarisation. The authors may consider removing this experiment from the manuscript. Figure 2—figure supplement 1 F-G-H presumably shows gated/sorted Foxp3+ cells: if this is the case, it needs to be clarified in the respective legend.

6. Figure 3B, and especially the meaning of the arrows, is unclear and may be omitted. Figure 3C shows substantial loss of Foxp3 gene expression with 6-AN treatment, but this does not appear to manifest in YFP expression as seen in Figure 3B. To check whether 6-AN affects iTreg polarization, the frequency of YFP+ cells and/or the gMFI of Foxp3-YFP should be assessed first in total CD4 T cells and not in gated YFP+ cells as apparently done here. In figure 3D-E, have YFP+ iTregs been sorted before assessing their suppressive function? In figure 3G, it is not clear what CONTROL means: is it the wild-type (not transgenic) mouse? The Treg gate looks very wide. Figure 3—figure supplement 1C is not mentioned in the text at the following point in the text: "higher infiltration of T cells (both CD4^+^ and CD8^+^ cells)". Regarding Figure 3—figure supplement 1L-M, are these cells MDSCs or rather pro-inflammatory neutrophils? Some more profiling is needed to identify these cells as MDSCs.

7. Reiterating an earlier point, the analysis of Figure 4 should be performed in Treg cells isolated from an inflammation-free system.

8. Statistical analysis is to be provided for data in Figures 3 and 4C

9. As a more general point, it should be clarified that data (and supporting bibliography) refer not to conventional T cells polarising into T helper subsets but to Tregs becoming unstable and gaining T helper-like features, which is a defined process.

10. The precise mechanism through which 6PGD destabilizes Tregs may be epigenetic (as mentioned in the discussion) as also metabolic (for instance related to a higher glucose capture). See for instance the two recent papers 10.1038/s41586-021-03326-4 and 10.1038/s41586-020-03045-2. The discussion could be integrated with some more hypotheses and speculations on the underlying mechanisms.

11. The manuscript would benefit from careful editing since several statements are not adequately supported by the data or by the quoted references.

*Reviewer #1:*

The authors employ chemical inhibition as well as genetic approaches to block 6PGD activity and report striking phenotypes of loss of immune-suppression in mice carrying a Treg-specific deletion. These phenotypes, which are characterized in considerable detail support an important role for the enzyme in Treg activation and homeostasis. The study includes description of disease phenotypes and cellular properties, which is supported by transcriptional analysis and detailed metabolite analysis at the cellular level. Mechanistic insights underpinning these outcomes and the regulation of 6PGD in the Treg lineage, however, remain to be addressed.

*Reviewer #2:*

This paper demonstrate that 6PGD, a rate-limiting enzyme in oxidative pentose-phosphate pathway (PPP), modulates Treg metabolism and functions. Indeed, mice with a Treg-restricted 6PGD deficiency (either constitutive or inducible) spontaneously develop a multi-organ scurfy-like lethal disease, characterised by less suppressive and unstable Tregs, which also have a skewed metabolism toward glycolysis, Krebs cycle and non-oxidative PPP.

The main strengths of the manuscript are (I) the original identification of a role for PPP in immune regulation, (ii) the high quantity and quality of data, spanning through several mouse models and validated through several methodologies (flow cytometry, functional assays, RNASeq, biochemistry, metabolic tracing, etc).

The precise mechanism through which 6PGD destabilises Tregs has not been fully elucidated here, and it may be epigenetic (as mentioned in the discussion) as well as metabolic (for instance related to a higher glucose capture, see for instance the two recent papers 10.1038/s41586-021-03326-4 and 10.1038/s41586-020-03045-2): the discussion could be integrated with some more hypotheses and speculations on the underlying mechanisms. The manuscript would also benefit from a careful revision, since several statements are not sufficiently supported by the data or by the quoted references.

*Reviewer #3:*

Mice with Treg cell specific 6PGD deficiency develop a severe lymphoproliferative disease and succumb to early death, suggesting a loss of Treg cell immunosuppressive function. Treg cell frequency in these mice is apparently only slightly reduced and is mostly affected by a transcriptional reprogramming to resemble that of conventional T effector subsets. The core ideas of this manuscript have a lot of potential in the area of Treg immunometabolism, which currently lags behind conventional CD4 and CD8 literature.

[Editors' note: further revisions were suggested prior to acceptance, as described below.]

Thank you for submitting your article "6-Phosphogluconate dehydrogenase (6PGD) a key checkpoint in reprogramming of Treg metabolism and function" for consideration by *eLife*. Your article has been reviewed by 3 peer reviewers, including Apurva Sarin as the Reviewing Editor and Reviewer #1, and the evaluation has been overseen by Satyajit Rath as the Senior Editor.

Essential revisions:

The revised manuscript addresses most earlier comments. The following essential modifications are required in the description of the results to address issues remaining in the study.

1. The early onset of systemic inflammation in mice with an ablation of 6PGD may modulate the analysis of cell intrinsic 6PGD dependencies, a limitation not ruled out in the study. This may be indicated in the description of results linked to Figure 2—figure supplement 1 and Figure 4 (metabolic reprogramming).

2. As indicated in reference PMID: 2738173, the phenotypic analysis may not be sufficient to identify cells as MDSCs. Since this is a not a typical context of MDSC expansion, such as cancer, but of lung inflammation, neutrophils may have also been recruited.

3. In experiments utilizing in vitro generated (induced) Tregs (iTregs), this must be explicitly stated in the description of the experiment in the text and accompanying legends.

---

## [Author Response]

Essential revisions:1. Most experiments describing the Treg cell phenotype and function are in Foxp3Cre 6PGDfl/fl mice, which has significant systemic inflammation. This confounds Treg cell phenotypes that are presented (e.g., RNA-seq profile, in vitro suppression, ECAR, OCR, adoptive transfer). The Foxp3Cre 6PGDfl/fl system is appropriate for the in vivo profiling data of conventional CD4, CD8, and Treg cells, but not to conclude direct effects of 6PGD loss within Treg cells. The characterization of 6PGD deficient Treg cells should be undertaken in an inflammation-free system, using either mixed bone marrow chimera mice or an inducible-cre system (see the following comment). Alternatively, key analyses may be performed using Foxp3Cre/+ heterozygous females, only if they do not develop inflammation. Characterization of the Treg phenotype in an inflammation-free background is critical to support the main conclusion of the study.

Response to the comment consists of two parts: (1) we evaluated the key observations of Tregs in inflammation free condition. (2) The goal of current paper is reporting observed phenotype and function and we aim to address the mechanism of action in future studies.

The main goal of current manuscript is to demonstrate the key role of 6PGD (the key enzyme in oxidative PPP) in modulating Treg function. We understand that there are questions unanswered, notably the mechanism of action and possible effects of inflammatory condition on the observed phenotype. However, to fully answer these questions is beyond the current scope. We have highlighted “inflammatory condition” throughout the manuscript to keep in mind that targeting 6PGD in Tregs comes along with systemic inflammation.

Suffice to say, we have examined the key findings in inflammation free conditions:

– We have generated Tregs from wild type (WT) naïve CD4^+^ T cells in the presence of 6PGD inhibitor (6-AN) and showed key findings as alteration of FoxP3/GATA3 expression, cytokine production shift and altered suppressive function (Figure 3A-E).

– We also showed comparable results in Tregs function (Figure 3H-O) using inducible FoxP3 mice (please see further details in response to comment 2) that provide inflammation free condition and the same inflammation baseline for both WT and 6PGD blocked Tregs.

– Furthermore, to evaluate metabolic shifts in Tregs in an inflammation free condition, we generated Tregs from naïve CD4^+^ T cells within 4 days in vitro (therefore there was no inflammatory environment) and tracked Glc and Gln metabolism which showed a similar pattern of shift as Tregs isolated from 6PGD^fl/fl^ FoxP3^Cre^ mice (please see added Figure 4 supplementary figure 2 and Figure 4 supplementary figure 3).

It should be noted that it is highly unlikely to generate inflammation free condition by Chimeric models of 6PGD deletion as it would need several weeks to develop chimera, while even 3 weeks of 6PGD blockade induces massive inflammation that kill the mice after birth. This is the reason why we took the in vitro approach. However, per the reviewer’s suggestion, the inducible FoxP3 Cre (tamoxifen-inducible Foxp3-CreERT2) is useful as we examined (Please see response to Comment 2).

2. The authors are advised to reassess the experiments using tamoxifen-inducible Foxp3-CreERT2 as the intricacies of this system appear to have been overlooked. Tamoxifen treatment does not induce Foxp3-EGFP expression in these mice as Foxp3-EGFP is expressed irrespective of tamoxifen (doi: 10.1126/science.1191996). Tamoxifen treatment will induce Cre-mediated gene deletion in Foxp3 expressing cells, but the cells are already EGFP+. In order to observe these cells, the mice must be crossed with a Rosa26-driven, stop-floxed fluorescent reporter mouse (e.g., YFP, available at jax.org/strain/006148), which will have fluorescent reporter expression only in the cells with successful Cre activation. The bulk EGFP+ cells contain a mixture of 6PGD+/+ and 6PGD-/- Treg cells. Furthermore, Foxp3-CreERT2 is a notoriously inefficient system for successful Cre activation, with only 50% or less of Treg cells actually having Cre-mediated gene deletion following tamoxifen treatment in vivo. If there are constraints on completing the experiment with the reporter as recommended, the data in its current form may be omitted from the study.

We have adopted the tamoxifen-inducible Foxp3-CreERT2 model for the “proof of concept” study, especially as a model for an inflammation free condition. Although the tamoxifen induction does not induce deletion in 100% of Tregs (possibly 50% based on the mentioned reference), it was sufficient to reduce tumor growth (i.e. reduced Treg suppressive function in vivo) and suppressive capacity in suppression assay in vitro. This is important in the context of Comment 1 discussion. The mentioned reference has very nicely shown that gene deletion (50%) only happened upon Tamoxifen induction, supporting “target specificity” and “inflammation free condition at baseline”. Our data indicate that 6PGD did not need to be deleted in 100% of Tregs to show significant phenotype. So, while we have not done the suggested improved experiment, the current data support our goal of defining 6PGD’s role in modulating Tregs function. We have added a statement in the result section to reflect the incomplete induction of FOXP3 deletion in the model used.

3. In the Results section, the sentence "As our preliminary studies showed that 6PGD but not G6PD blockade induces T cells activation (Figure 1—figure supplement 1 B-C). " is truncated. The preliminary data shown in suppl. Figure 1B-C indicates that a 6PGD pharmacological inhibitor enhances IFN-γ production by in vitro activated CD4 T cells, not that it "induces" T cell activation: the text should be corrected accordingly. In supplemental figure 1, the effects of DHEA on cell viability are noticeable. Any activation defects on the cells are likely a secondary consequence. Is it possible to use a lower dose to avoid so much cell death? In Figure 1G, the sentence "Lymphoproliferation was attributed to a higher number of CD4^+^ and CD8^+^ T cells in spleen/lymphoid organs should be corrected, since this figure shows frequencies and not counts. Further, Figure 1G only shows one plot, but it is from multiple organs according to the text. Are the other plots missing?

The sentence was corrected based on the comment to show enhanced IFN-γ production and not “induction of activation” in T cells.

The main point of supplemental figure 1B-C was to show noticeable effect of DHEA on activated T cells viability. While the same concentration of 6-AN and DHEA inhibitors (10 µM) were used, DHEA instigated poor T cells viability but not 6-AN. This shows the critical role of G6PD in cells viability and therefore its targeting is not tolerable by T cells. Meanwhile, blocking 6PGD was tolerable by the activated T cells and resulted in higher IFN-g production presumably by the induced metabolic shifts. This preliminary experiment provided the reason why we investigated 6PGD’s role in T cell activation. Text on page 4 was revised to substantiate this point.

For Figure 1G, the word “Count” was corrected for “Frequency”. (Count per organ is presented in Figure 1E). We have evaluated both lymph nodes and spleens for markers mentioned in Figure 1G-K. As both spleen and lymphnodes show similar pattern or responses, we presented splenocytes data as an example. We have revised the relevant Results text on page 5 accordingly. Moreover, the enlarged spleen and lymph node size shown in Figure 1D suggests lymphoproliferation in both spleen and lymph nodes.

4. Regarding Treg frequency in Figure 2A, it is more informative to show the percentage of Foxp3+ in gated CD4 T cells, and also to show the gMFI of Foxp3 in gated Foxp3+ cells (it seems much lower in fl/fl mice; indeed this is confirmed by RT-PCR in Figure 2E). Is the difference between the two groups statistically significant? Statistics for frequencies and total numbers of Treg cells are missing. It is unclear why iTreg cells were used for figure 2C? A 50% reduction in Foxp3 gene expression, (Figure 2E), is substantial and seems incongruent with the claim of only a "slight decrease" in YFP MFI. Does direct Foxp3 staining and MFI quantification show the same result?

We added the percentage of Foxp3+ in gated CD4 T cells in Figure 2A. We also added Figure 2B to show gMFI of Foxp3 in gated Foxp3+ cells. Statistical analysis for the two panels are provided to show significantly lower frequency of FoxP3+ T cells in 6PGD^fl/fl^ FoxP3^Cre^ group. Figure 2C (original) was eliminated to prevent confusion.

Decreased FoxP3 expression (Figure 2E) is consistent with lower levels of Foxp3 staining in CD4^+^ cells and gMFI quantifications (Figure 2A-B) in 6PGD blocked Tregs. This result was also reproduced when Tregs were generated with 6PGD inhibitor 6-AN, in vitro (Figure 3C). About 50% decrease in YFP+ cells is also evident in 6PGDfl/fl FoxP3Cre mice (Data not shown in paper; Please see Author response image 1) Please note that Figure 2D is representative of “Sorted YFP+ cells” that have been used in all YFP+ sorted experiment (e.g. Figure 2E-G and Figure 2—figure supplement 1 and Figure 4 and supplements). We revised the manuscript text on page 6 to clarify this.

**Author response image 1. sa2fig1:** YFP expression in CD4^+^ gated cells from mice spleen.

5. Figure 2—figure supplement 1D does not explain Treg plasticity since it refers to conventional effector T cell polarisation. The authors may consider removing this experiment from the manuscript. Figure 2—figure supplement 1 F-G-H presumably shows gated/sorted Foxp3+ cells: if this is the case, it needs to be clarified in the respective legend.

We removed the Figure 2—figure supplement 1D. In case of Figure 2—figure supplement 1 F-G-H (original and E-F-G of revised figure) graphs show “Splenic CD4^+^ T cells” as mentioned in the figure legends. While 6PGD blocked Tregs show altered markers of CD4^+^ T subsets (revised Figure 2—figure supplement 1A-C and also Figure 2G and Figure 3C), E-G of revised figure 2—figure supplement 1 provide additional information regarding CD4^+^ T subsets and indicate general shift of CD4^+^ T cells toward Th1, Th2 and Th17 cells subsets in 6PGD^fl/fl^ FoxP3^Cre^ mice. Text was clarified in page 7.

6. Figure 3B, and especially the meaning of the arrows, is unclear and may be omitted. Figure 3C shows substantial loss of Foxp3 gene expression with 6-AN treatment, but this does not appear to manifest in YFP expression as seen in Figure 3B. To check whether 6-AN affects iTreg polarization, the frequency of YFP+ cells and/or the gMFI of Foxp3-YFP should be assessed first in total CD4 T cells and not in gated YFP+ cells as apparently done here. In figure 3D-E, have YFP+ iTregs been sorted before assessing their suppressive function? In figure 3G, it is not clear what CONTROL means: is it the wild-type (not transgenic) mouse? The Treg gate looks very wide. Figure 3—figure supplement 1C is not mentioned in the text at the following point in the text: "higher infiltration of T cells (both CD4^+^ and CD8^+^ cells)". Regarding Figure 3—figure supplement 1L-M, are these cells MDSCs or rather pro-inflammatory neutrophils? Some more profiling is needed to identify these cells as MDSCs.

We believe that the reviewer meant arrows in the schemes of Figure 3A (original) being unclear (as the Figure 3B was a flowcytometry graph). We have removed the figure 3A panel.

The original Figure 3B (now Figure 3A) shows the percentage of YFP+ cells. The YFP gMFI (gated on CD4^+^ cells) for respective treatments was determined and presented in bar graph as new Figure 3B. Lower YFP gMFI in 6-AN treated cells is akin to lower FoxP3 mRNA levels detected by Real-time PCR (Figure 3C).

In figure 3D-E, YFP+ iTregs have been sorted before assessing their suppressive function. We clarified this in the figure legend.

In figure 3G, “Control” is the wild type mice. These cells are used to determine autofluorescence baseline for eGFP expression. Graph label was changed from “Control (EGFP-)” to “WT (EGFP-)”. Gating was done based on the WT baseline.

Text was corrected on page 9 to cite Figure 3—figure supplement 1C after reporting "higher infiltration of T cells (both CD4^+^ and CD8^+^ cells) in the lung.

Figure 3—figure supplement 1L-M illustrated MDSCs accumulation in inflamed lungs, which is an additional marker along with other results to show uncontrolled allergic responses in lung of mice with 6PGD deletion in their Tregs. We used CD11b+Ly6G+ cells as marker of PMN MDSCs (ref to: PMID: 27381735 and PMID: 32549755). To investigate details of MDSCs properties upon 6PGD blockade is beyond the scope of this study.

7. Reiterating an earlier point, the analysis of Figure 4 should be performed in Treg cells isolated from an inflammation-free system.

To analyze Tergs metabolism in an inflammation-free system, we generated Tregs in vitro for the tracing study. Results are presented in Figure 4—figure supplement 2 and Figure 4—figure supplement 3. These results indicate a similar trend of metabolic shifts in 6PGD deficient Tregs isolated form 6PGD^fl/fl^ FoxP3^Cre^ mice and 6PGD deleted Tregs generated in vitro in the inflammation free environment.

8. Statistical analysis is to be provided for data in Figures 3 and 4C

Statistical analysis was done for the mentioned figures and bar graphs provided.

9. As a more general point, it should be clarified that data (and supporting bibliography) refer not to conventional T cells polarising into T helper subsets but to Tregs becoming unstable and gaining T helper-like features, which is a defined process.

Text was corrected on page 26 for the mentioned point.

10. The precise mechanism through which 6PGD destabilizes Tregs may be epigenetic (as mentioned in the discussion) as also metabolic (for instance related to a higher glucose capture). See for instance the two recent papers 10.1038/s41586-021-03326-4 and 10.1038/s41586-020-03045-2. The discussion could be integrated with some more hypotheses and speculations on the underlying mechanisms.

Discussion on page 14 was revised to discuss more potential mechanisms also referring the above references.

11. The manuscript would benefit from careful editing since several statements are not adequately supported by the data or by the quoted references.

We revised the text carefully to be sure that is representative of our results or the literature.

[Editors' note: further revisions were suggested prior to acceptance, as described below.]

Essential revisions:The revised manuscript addresses most earlier comments. The following essential modifications are required in the description of the results to address issues remaining in the study.1. The early onset of systemic inflammation in mice with an ablation of 6PGD may modulate the analysis of cell intrinsic 6PGD dependencies, a limitation not ruled out in the study. This may be indicated in the description of results linked to Figure 2—figure supplement 1 and Figure 4 (metabolic reprogramming).

Text was modified to point out the limitation for the findings on Tregs isolated from pro-inflammatory mice. This include page 6 and 7 for Figure 2—figure supplement 1 and page 10 for Figure 4 (marked by track change).

2. As indicated in reference PMID: 2738173, the phenotypic analysis may not be sufficient to identify cells as MDSCs. Since this is a not a typical context of MDSC expansion, such as cancer, but of lung inflammation, neutrophils may have also been recruited.

We removed the “MDSC” from the text and figure legend and indicated only CD11b+ Ly6G+ cells along with the mentioned reference (page 10 and Figure 3—figure supplement 1 legends). These text changes were made as we did not delineate CD11b+ Ly6G+ cells to be MDSC or neutrophils, which is beyond the scope of current study.

3. In experiments utilizing in vitro generated (induced) Tregs (iTregs), this must be explicitly stated in the description of the experiment in the text and accompanying legends.

We clarified the text and figure legends for the Tregs generation in vitro, anywhere describing these cells. This includes, figure 3, Figure 4—figure supplement 2 and Figure 4—figure supplement 3 (text and figure legends).